



# On the lidar-turbulence paradox and possible countermeasures

Alfredo Peña[1], Ginka G. Yankova[1], and Vasiliki Mallini[1]

[1]DTU Wind and Energy Systems, Technical University of Denmark, Roskilde, Denmark

**Correspondence:** Alfredo Peña (aldi@dtu.dk)

**Abstract.** We describe the major difficulties in establishing a physics-based method that corrects lidar-based turbulence measures so that they become equivalent to standard turbulence measures. The difficulties encompass the so-called lidar-turbulence paradox, which we circumvent in two ways. The first uses a physics-based lidar-turbulence model and the second directly uses lidar measurements, both approaches aiming at training neural networks. The measurements are from continuous-wave Doppler

lidar wind profilers deployed besides a tall 250-m meteorological mast at the Østerild test station in Denmark. Sonic anemometers on the mast match four lidar measurement levels, from 37 up to 241-m height. The physics-based lidar-turbulence model predicts well the behavior of the ratio of the lidar-to-sonic along- and cross-wind velocity variance up to 103 m. However, it predicts further decreasing ratios at 175 and 241 m, while the observations show increasing ratios for a number of stability conditions and length-scale ranges. The physics-based lidar-turbulence model is used to produce physics-based datasets, which

are utilized to train neural networks. Compared to turbulence intensity measurements from a first lidar, the predictions of these neural networks are in better agreement with the sonic-based measures for most mean wind speed bins at 37 and 103 m. At 175 and 241 m, the predictions' accuracy reduces and better agreement is achieved within the highest mean wind speed ranges only. Measurements from a second lidar are used to generate predictions of turbulence intensity with neural networks trained with measurements from the first lidar. At 37 and 103 m, these predictions are also in better agreement with the sonic-based measures than those of the second lidar for most mean wind speed ranges. However, at 175 and 241 m, turbulence measures

derived from the second lidar are generally close to the sonic-based values, while the predictions overestimate them. We speculate either that the assumption of turbulence homogeneity within the lidar scanning pattern might not hold at the site and/or that the Doppler radial velocity spectra of the lidars might be contaminated, thus impacting the radial velocity estimates particularly with increasing focus distance.

## 1   Introduction

The wind energy community is eager to establish a methodology to correct Doppler wind lidar turbulence measurements (hereafter lidars as this study concentrates only on these types of lidars) so that lidar-based turbulence measures become closer or equivalent to turbulence measures from standard (and standarized) anemometers such as sonic and cup anemometers (Clifton et al., 2018; Goit et al., 2019). This is mainly because lidars are versatile and affordable and can accurately measure winds

within and beyond the limits of meteorological masts (Floors et al., 2015; Filioglou et al., 2022). In particular, for offshore applications, floating lidars are nowadays the standard for assessing wind resources, as the deployment and maintenance of





instruments on tall meteorological towers offshore is currently too expensive (Gutiérrez-Antuñano et al., 2018). With regard to versatility, lidars are used for different applications, e.g. wake studies (Doubrawa et al., 2019), determining inflow conditions (Fu et al., 2023), power performance analyses (Sebastiani et al., 2023), and wind turbine control (Schlipf et al., 2015). The

wind energy community is therefore making efforts in the development of recommended practices and standards to enhance the adoption of measurements from ground-based, nacelle-based, and floating lidars (Clifton et al., 2018).

Corrections for lidar-based turbulence measures have been investigated for decades (Eberhard et al., 1989; Frehlich, 1997; Smalikho et al., 2005; Newman and Clifton, 2017). The major difficulties for establishing a correction are related to two points: turbulence (cross-)contamination and turbulence filtering due to probe-volume averaging. Here, we briefly introduce these two

points, which leads to what we later refer to as the lidar-turbulence paradox.

### 1.1 Turbulence contamination

To reconstruct the velocity components, most lidars scan the atmosphere at different positions in space, since in most cases only one lidar unit is used for the task. Such scans complicate the computation of velocity-component turbulence measures because a lidar beam only measures the line-of-sight velocity, which is the radial velocity $v_r$ along the beam direction. In most

cases, scanning at several positions leads to what is commonly referred to as 'contamination' of the target velocity-component turbulence measure because, e.g., velocity-component variances become dependent on other components of the turbulence covariance tensor. To briefly illustrate the contamination of velocity variances, we first study a lidar scanning with a single beam at an elevation angle $\phi$ (see Fig. 1).

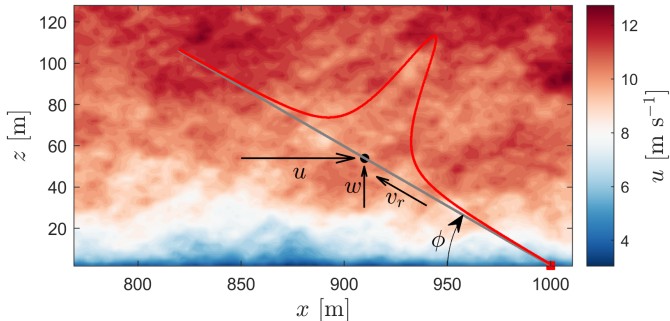

**Figure 1.** A lidar (red rectangle) scanning the turbulent atmosphere at an elevation $\phi$ with one beam focused at a point (black circle). A probe length typical of a continuous-wave lidar is shown in the red curve. See text for further details

We define the wind vector $\boldsymbol{v} = (u, v, w)$ using a right-handed Cartesian coordinate system, where $u$ is aligned with the mean

wind, $v$ is transverse to $u$ (so $v$ has zero mean), and $w$ is perpendicular to these two, with $u$, $v$, $w$ aligned with the directions $x$, $y$, $z$, respectively. Assuming that the lidar beam is both parallel to the mean wind and measures over a point (and not over a volume), the beam radial velocity for the lidar in Fig. 1 is $v_r = u \cos\phi - w \sin\phi$, whose variance is (Eberhard et al., 1989)

$$\sigma_{v_r}^2 = \sigma_u^2 \cos^2\phi + \sigma_w^2 \sin^2\phi - 2\langle u'w' \rangle \cos\phi \sin\phi, \tag{1}$$





where $\langle u'w' \rangle$ is the $uw$-covariance. For positive $\phi$ values, $\langle u'w' \rangle$ positively contaminates $\sigma_{v_r}^2$ because the term $\langle u'w' \rangle$ is negative under positive vertical velocity shears, which is generally the case of the atmospheric surface-layer flow. Further, depending on both the $\phi$ value and the ratios of velocity variances and covariances, $\sigma_{v_r}^2$ can be larger than $\sigma_u^2$, which is the velocity variance that most people target within the wind energy community. Assuming, e.g., the ratio $\sigma_w^2/\sigma_u^2 = 0.25$ (IEC, 2019) and $\langle u'w' \rangle/\sigma_u^2 \approx -1/6$ (Stull, 1988), Eqn. (1) becomes $\sigma_{v_r}^2 = \sigma_u^2 \cos^2 \phi + (1/4)\sigma_u^2 \sin^2 \phi + (1/3)\sigma_u^2 \cos\phi\sin\phi$, which for the case $\phi = 30°$ results in $\sigma_{v_r}^2 = 0.96\sigma_u^2$, whereas for the case $\phi = 10°$, $\sigma_{v_r}^2 = 1.03\sigma_u^2$. For cases with negative $\phi$ values, the $uw$-covariance negatively contaminates $\sigma_{v_r}^2$ and so $\sigma_{v_r}^2 < \sigma_u^2$. In the previous example, the lidar scans at one position only and the contamination is rather simple. When reconstructing velocity components from scans at different positions, cross-contamination can also take place, leading to resonance effects on the turbulence spectra of the target velocity component (Sathe et al., 2011; Kelberlau and Mann, 2019, 2020).

## 1.2 Probe-volume averaging

Lidars measure within a probe volume (see Fig. 1), and depending on the lidar type and on the measurement range for the case of continuous-wave (CW) lidars, the length of this probe volume determines the size of the atmospheric turbulent eddies that we can measure. The probe volume therefore acts as a turbulence filter, also known as probe-volume averaging effect. In principle, we could only perform 'perfect' measurements of turbulence using lidars if we had three units and their probe volumes were small enough to measure the eddies with most energy.

An interesting part of the problem is what we here refer to as the lidar-turbulence 'paradox'. To understand the reasoning behind the paradox, we can start by looking at the expression for the radial velocity spectrum of a lidar beam, which is given as (Mann et al., 2009):

$$F_{v_r}(k_1) = n_i n_j \iint |\hat{\varphi}(\boldsymbol{k} \cdot \boldsymbol{n})|^2 \Phi_{ij}(\boldsymbol{k}) dk_2 dk_3, \tag{2}$$

where $\hat{\varphi}$ is the Fourier transform of the weighting function $\varphi$ that describes the probe-volume characteristics, $n_{i,j}$ are the components of the unit vector of the lidar beams, $k_{2,3}$ the wave numbers in the transverse and vertical directions ($k_1$ is on the along-wind direction), and $\Phi_{ij}$ is the spectral velocity tensor.

The lidar turbulence paradox appears already in Eqn. (2): we measure turbulence with a lidar and in the easiest of the cases, e.g., when the lidar beam is aligned with $u$, i.e., $i, j = 1$, we only need to compute $\hat{\varphi}^2 \Phi_{11}$ as a function of the wave number. However, the components of $\Phi_{ij}$ depend on the turbulence characteristics; e.g., on the turbulent length scale (Kaimal and Finnigan, 1994). Therefore, for this 'simple' lidar configuration, to determine the effect of the probe volume on $F_{v_r}(k_1)$, and thus on $\sigma_{v_r}^2$, which is the integral of $F_{v_r}(k_1)$, we need to know in advance the characteristics of turbulence, which we are trying to determine in the first place with the lidar measurements. One can now imagine that when the lidar beam is not perfectly aligned with none of the three velocity components, as in most scanning configurations, the complexity of the problem increases because all components of the spectral tensor might contribute to the radial velocity spectrum and 'interplay' with the weighting function.





There are ways to avoid the paradox. If we assume that the Doppler spectrum of radial velocities measured by a lidar contains turbulence information only, we can use this Doppler spectrum to determine the so-called unfiltered radial velocity variances, i.e., radial velocity variances that do not suffer from probe-volume averaging effects. Using unfiltered radial velocity variances and depending on the scanning geometry, one can determine unfiltered velocity-component covariances. Using measurements from a CW wind profiler, Mann et al. (2010) computed $uw$-covariances using estimates of the unfiltered radial velocity variance of the two sets of lidar beams that were aligned with the mean wind direction. Also, if unfiltered radial velocity variances are available, we can in principle compute all six Reynolds stresses with one single unit measuring at minimum six positions. A procedure for computing all stresses is explained in Sathe et al. (2015) for a ground-based wind profiler and in Fu et al. (2023) for a nacelle-based lidar. Unfortunately, it is not common to store information with regard to the Doppler spectrum of radial velocities of each of the lidar beams. In some cases, we cannot access information on the radial velocity estimate; the rawest level of information available in a number of commercial lidars concerns the time series of reconstructed horizontal velocity and direction.

In summary, we deal with reconstructed velocity components, whose second-order moments are both 'contaminated' from contributions from the other components of the spectral tensor and 'filtered' due to probe-volume averaging effects. Note that, under certain scanning geometries, the probe-volume averaging effect can be compensated with positive turbulence (cross-)contamination, and lidar-based turbulence measures might thus appear accurate or close to those from standard anemometers. It is also important to mention that both effects are also present in measurements of other type of remote sensors, such as radars and sodars, which due to their nature generally measure over larger probe volumes and are less accurate than lidars. Here, we explore an alternative method, a lidar-turbulence paradox countermeasure that corrects lidar-based turbulence measurements. The method is based on training neural networks (NNs) with either physics-based datasets or directly with lidar measurements. The physics-based NNs allow us to study the sensitivity of lidar-derived turbulence measures on model-based turbulence characteristics, since the NNs can be trained with different sets of inputs and the NN framework is ideal to evaluate their level of importance.

The work is organized as follows. Section 2 complements the background on turbulence and lidars already introduced in Sect. 1. Section 3 introduces the methodologies that we use to correct the lidar turbulence measurements. In Sect. 4, we present the site and the lidar and meteorological mast measurements, which we use to evaluate different methods to correct the lidar turbulence measurements. The analysis of these measurements is presented in Sect. 5. Section 6 shows the results of the different models and methodologies. Sections 7 and 8 provide a discussion and the conclusions of the study.

## 2 Background

Section 1 briefly introduced some of the basic background to understand the lidar-turbulence challenge. Here, we complement this background by first introducing turbulence generalities (Sect. 2.1), second further illustrating the lidar-turbulence paradox (Sect. 2.2), and third introducing the physics-based lidar-turbulence model (Sect. 2.3).





## 2.1 Turbulence generalities

The spectral velocity tensor for homogeneous turbulence in Eqn. (2) is defined as

$$\Phi_{ij}(\boldsymbol{k}) = \frac{1}{(2\pi)^3} \int R_{ij}(\boldsymbol{r}) \exp(i\boldsymbol{k} \cdot \boldsymbol{r}) d\boldsymbol{k}, \tag{3}$$

where $R_{ij} = \langle v_i'(\boldsymbol{x}) v_j'(\boldsymbol{x}+\boldsymbol{r}) \rangle$ is the cross-variance function, $\boldsymbol{x}$ the position vector, and $\boldsymbol{r}$ the separation vector. The one-point spectra of the velocity components are thus:

$$F_{ij}(k_1) = \iint \Phi_{ij}(\boldsymbol{k}) dk_2 dk_3, \tag{4}$$

and the one-point velocity variances and covariances are then given as

$$\langle u_i' u_j' \rangle = \int F_{ij}(k_1) dk_1. \tag{5}$$

Equation (4) can be used to describe the velocity spectra that one computes from measurements using an ideal anemometer. Note that Eqn. (4) is a simplified version of Eqn. (2); hereafter we assume that the measurements of a sonic anemometer are close to ideal, i.e., $\hat{\varphi} \approx 1$. Here, we use the three-dimensional homogeneous turbulence spectral tensor model by Mann (1994) (hereafter Mann model) to describe $\Phi_{ij}$. The Mann model contains three parameters in addition to the wavenumber vector $\boldsymbol{k}$; the dissipation rate of turbulence $\alpha \epsilon^{2/3}$, the turbulence length scale $L$, and the turbulence anisotropy $\Gamma$. For a detailed analysis of the behavior of the Mann parameters in the atmosphere under a range of turbulence, atmospheric stability, wind speed conditions, and heights, we refer to Peña et al. (2010), Kelly (2018), and Peña (2019).

## 2.2 Lidar-turbulence paradox

We can now further examine the lidar-turbulence paradox introduced in Sect. 1.2 for the simple lidar configuration illustrated in Fig. 1 with the Mann model tensor, since it has a non-zero $\Phi_{13}$ component, i.e., it models the $uw$-covariance. Based on the description of $\Phi_{ij}$ by the Mann model, the ratios $\sigma_w^2/\sigma_u^2$ and $\langle u'w' \rangle/\sigma_u^2$, which can be computed with Eqns. (4) and (5) are highly dependent on $\Gamma$. For $\Gamma = 4$, $\sigma_w^2/\sigma_u^2 \approx 0.25$, which is similar to the recommended value of IEC (2019) and $\langle u'w' \rangle/\sigma_u^2 \approx -0.25$. For $\Gamma = 2$, which is a typical value of conditions close to isotropic, $\sigma_w^2/\sigma_u^2 \approx 0.61$ and $\langle u'w' \rangle/\sigma_u^2 \approx -0.28$. Therefore, when scanning at $\phi = 30°$, the beam variance $\sigma_{v_r}^2$ of this particular lidar beam geometry is always higher than the $u$-variance within the range $\Gamma = 2$–$4$ if probe-volume averaging effects are not accounted for, i.e., the results for the case $z_R = 0$ m in Fig. 2. The Rayleigh length $z_R$ characterizes the length of the probe volume of a CW lidar (Sonnenschein and Horrigan, 1971):

$$z_R = \frac{\lambda d_f^2}{\pi a_0^2}, \tag{6}$$

where $\lambda$ is the laser wavelength, $d_f$ the focus distance, and $a_0$ the effective beam radius at the output lens. For a CW lidar with a weighting function following a Lorentzian shape (red curve in Fig. 1), $\hat{\varphi}(\boldsymbol{k}) = \exp(-|\boldsymbol{k}|z_R)$.

Figure 2 shows that the ratio $\sigma_{v_r}^2/\sigma_u^2$ (squares) becomes highly dependent on $z_R$ and its relation to the Mann turbulent length scale $L$. For $z_R = 5$ m, the ratio $\sigma_{v_r}^2/\sigma_u^2$ drops and is always lower than one, with larger and lower turbulence filtering effects

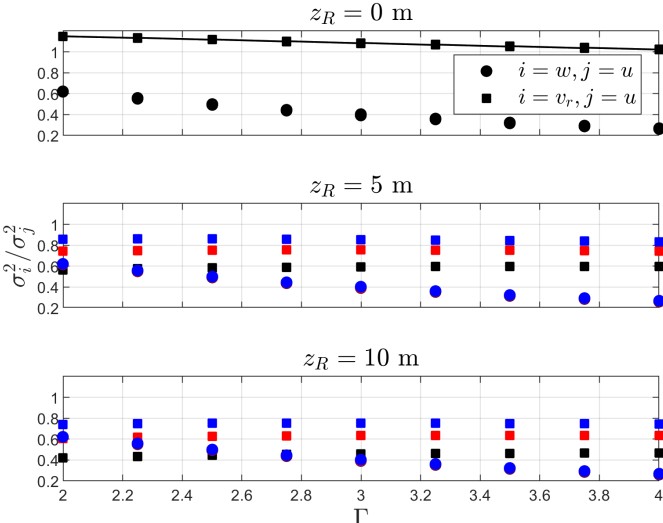

**Figure 2.** Ratio of $w$-to-$u$ (circles) and $v_r$-to-$u$ (squares) variances as a function of $\Gamma$ for the lidar in Fig. 1 scanning at $\phi = 30°$. Markers correspond to the variances based on Eqn. (5) and the integral of Eqn. (2). Solid line corresponds to Eqn. (1). For $z_R > 0$ m, markers correspond to $L = 10$ m (black), $L = 25$ m (red), and $L = 50$ m (blue)

under conditions with lower and higher $L$ values, respectively. For a relatively small probe length ($z_R = 10$ m), we observe that within a range of low values of $\Gamma$, $\sigma_{v_r}^2/\sigma_u^2 < \sigma_w^2/\sigma_u^2$, i.e., the beam variance is lower than the variance of $w$. Turbulence contamination and probe-volume effects are therefore mixed up and the desired correction to convert $\sigma_{v_r}^2$ into $\sigma_u^2$, which is a

145 (the) key turbulence measure in wind energy, becomes at the very least dependent on both $\Gamma$ and $z_R/L$. Thus, in principle and paradoxically, we need to know the turbulence characteristics to determine how much turbulence filtering and contamination result from the lidar scans.

### 2.3 The physics-based lidar-turbulence model

Since we evaluate possible lidar-turbulence countermeasures using measurements from ground-based conical scanning lidars,
we use the physics-based lidar-turbulence model that was already described by Sathe et al. (2011). They derived expressions for the spectra of the reconstructed velocities measured by this type of lidar configuration. The vertical velocity spectra of this lidar (hereafter $l$ refers to lidar) are given by:

$$F_w^l(k_1) = \left( \frac{1}{\cos^2 \psi} \right) \widehat{T_f}(k_1) \iint \Phi_{ij}(\boldsymbol{k}) \alpha_i(\boldsymbol{k}) \alpha_j^*(\boldsymbol{k}) dk_2 dk_3, \tag{7}$$

where $\alpha$ is a spectral weighting function ($^*$ means complex conjugation), $\psi$ the lidar half-opening angle, and $\widehat{T_f}$ a spectral
transfer function accounting for the low-pass filter effect due to the time the lidar takes to scan the cone. For the $u$ and $v$ velocities, the spectrum is similar to that in Eqn. (7) but the term $\cos^2 \psi$ in the denominator should be replaced by $\sin^2 \psi$ and





the weighting function $\alpha$ by $\beta$ and $\gamma$, respectively. For a CW lidar, these weighting functions are

$$\alpha_i(\boldsymbol{k}) \quad = \quad \frac{1}{2\pi}\int_0^{2\pi} n_i(\theta)\exp\left(\mathrm{i}d_f\boldsymbol{k}\cdot\boldsymbol{n}(\theta)\right)\exp\left(-z_R|\boldsymbol{k}\cdot\boldsymbol{n}(\theta)|\right)d\theta, \tag{8}$$

$$\beta_i(\boldsymbol{k}) \quad = \quad \frac{1}{\pi}\int_0^{2\pi} \cos\theta\, n_i(\theta)\exp\left(\mathrm{i}d_f\boldsymbol{k}\cdot\boldsymbol{n}(\theta)\right)\exp\left(-z_R|\boldsymbol{k}\cdot\boldsymbol{n}(\theta)|\right)d\theta, \tag{9}$$

$$\gamma_i(\boldsymbol{k}) \quad = \quad \frac{1}{\pi}\int_0^{2\pi} \sin\theta\, n_i(\theta)\exp\left(\mathrm{i}d_f\boldsymbol{k}\cdot\boldsymbol{n}(\theta)\right)\exp\left(-z_R|\boldsymbol{k}\cdot\boldsymbol{n}(\theta)|\right)d\theta, \tag{10}$$

where $\boldsymbol{n}(\phi,\theta) = (\cos\theta\sin\psi, \sin\theta\sin\psi, \cos\psi)$ is the unit vector describing the lidar scanning pattern with $\theta$ being the azimuthal positions. By combining Eqns. (6)–(10), we can compute the velocity spectra, whose integrals (see Eqn. 5) result in the lidar-reconstructed velocity variances.

## 3 Methods

The following list describes the number of methods and analyses that we use to evaluate the abilities of the models to convert lidar-to-sonic turbulence measures.

1. We use measurements from a ground-based CW lidar (lidar 1) and from sonic anemometers on a tall meteorological mast to evaluate the ability of the physics-based lidar-turbulence model in Sect. 2.3 to reproduce lidar-to-sonic turbulence measures. The evaluation (see Sect. 6.1) is performed using lidar and sonic-anemometer observations, which are both described and classified in Sect. 5.

2. We also use the physics-based lidar-turbulence model (Sect. 2.3) to train physics-based NNs (PBNNs), which are described in Sect. 3.1. The PBNNs are first cross-validated against simulated data derived from the physics-based lidar-turbulence model itself (Sect. 6.2.1). The PBNNs can therefore be seen as a multidimensional look-up-table (LUT). The assessment of the PBNNs is performed by using a K-fold cross-validation. We test multiple PBNNs, which are derived from different combinations of inputs to the NNs. This approach allows us to examine the significance of the predictors (inputs).

3. We evaluate PBNNs (trained similarly as for the cross-validation above) against the full set of measurements from lidar 1 (Sect. 6.2.2). Here, the inputs correspond to combinations of possible lidar-measured parameters. We also use the output of a PBNN to evaluate turbulence intensity (TI) lidar-based predictions with those from the sonic and original lidar measurements for a number of wind speed ranges.

4. We use measurements from lidar 1 to train data-driven NNs (DDNNs), which are described in Sect. 3.2. The DDNNs are first evaluated against lidar 1 measurements using a K-fold cross-validation (Sect. 6.3.1). We also test multiple DDNNs, which are derived from different combinations of lidar-measured inputs to the NNs and study their importance.





5. We train full DDNNs with the lidar 1 data using the combination of inputs that give the best performance when cross-validating. We then evaluate the DDNNs using measurements from another and independent ground-based CW lidar (lidar 2) in Sect. 6.3.2.

## 3.1 The physics-based neural networks

The physics-based lidar-turbulence model in Sect. 2.3 is computationally expensive and paradoxical as it needs information on turbulence itself to correct the lidar-turbulence measures. We therefore construct datasets of outputs from the physics-based lidar-turbulence model. We construct four of them, since lidar turbulence is focus-distance and, thus, height dependent, and as shown later in Sect. 4, there are four matching vertical levels between the lidar and sonic-anemometer measurements. Each of these datasets is constructed such that the output of the physics-based lidar-turbulence model covers the possibly broad range of lidar turbulence measurements, which are used to predict the sonic-equivalent turbulence measurements. Therefore, we use information from the lidar measurements as proxy to establish a range of Mann parameters that cover the turbulence 'climatology' of the given period of observations. The procedure for constructing each of these datasets follows these steps:

1. We use all 10-min samples ($m$) from lidar 1 observations to determine the observed turbulence ranges. Specifically, we use the lidar-reconstructed $u$-variance $\sigma_{u_l}^2$ to derive a range of $\alpha\epsilon^{2/3}$ values assuming $\alpha\epsilon^{2/3} \approx 7.5\sigma_{u_l}^2/z^{2/3}$.

2. For all $m$ 10-min samples, we estimate the Mann turbulence length scale $L$ using the expression from Kelly (2018) adapted to the lidar measures, i.e., $L \approx \sigma_{u_l}/(dU/dz)$. As shown in Sect. 5, the lidar-based mean wind speed agrees very well with that of the sonic anemometers at the matching heights and so we assume that the vertical wind speed gradient $dU/dz$ based on lidar measurements is equivalent to the one based on sonic measurements. We use the sonic-based vertical wind speed gradient because the mast includes a sonic measurement at a lower height than the lowest lidar-sonic matching height. This allows us determining more accurately the vertical wind speed gradient at all the lidar-sonic matching levels.

3. We only lack information about $\Gamma$: from velocity spectra analysis based on the sonic anemometer data of the Østerild mast (Peña, 2019), $\Gamma = 1.5$–$4.0$ covered a broad range of atmospheric turbulence conditions. An additional preliminary dataset of size $m \times 6$[1] that combines $\alpha\epsilon^{2/3}$, $L$, and $\Gamma$ values is constructed to pre-compute theoretical sonic-based velocity variances. This is a much faster procedure than the computation of lidar-derived velocity variances with Eqns. (7)–(10), since we can use a two-parameter precomputed LUT of Mann-based velocity spectra (Eqn. 4). The procedure is explained in Peña et al. (2017) and uses the identity:

$$F_{ij}(k_1; \alpha\varepsilon^{2/3}, L, \Gamma) = L^{5/3}\alpha\varepsilon^{2/3}F_{ij}(k_1 L; 1, 1, \Gamma). \tag{11}$$

4. We randomly select 1000 out of the $m \times 6$ possible atmospheric states, i.e., 1000 combinations of Mann parameters. We only select 1000 states because of the cost of lidar-based turbulence calculations, i.e., Eqns. (7)–(10). The final

---

[1]6 from varying $\Gamma = (1.5 : 0.5 : 4.0)$





four datasets are constructed with 1000 samples each and include the theoretical lidar-based and sonic-based velocity
variances.

We are now ready to train PBNNs. It is beyond the scope of the study to test different types and characteristics of NNs, e.g., the number of hidden layers or neurons. However, the first trials, which were performed only on physics-based datasets, revealed that NNs were superior to a number of decision trees tested using different algorithms for classification and regression. From these first trials, the performance of the NNs did not increase using more than 20 neurons or using more complex
layer structures. Also, of the two algorithms tested, Levenberg-Marquardt and Bayesian regularization, the latter was superior. We therefore use the default NN `fitnet` for function fitting problems of Matlab (The MathWorks Inc., 2022). This is a feedforward NN and we use an architecture comprising one hidden layer with 20 neurons and one output neuron because we only need one response value: the sonic-equivalent $u$-variance.

### 3.1.1 Cross-validation

PBNNs are cross-validated using the model-based datasets constructed in Sect. 3.1. We cross-validate multiple PBNNs, whose sole difference is the inputs they use, allowing us to study their importance. The assessment of the model is performed using a K-fold cross-validation. The 1000-sample model-based datasets are randomly permuted and subdivided into a training and testing subset, with 90% of the datasets going to training and 10% to testing. We train NNs with the training subset, generate predictions using different inputs of the testing subset, and, finally, compare the NN-based predictions with the known value of
the prediction, in this case the modeled sonic-based $u$-variance. We compute the root mean square error (rmse), mean absolute percentage error (mape), and the slope of a linear fit through origin between the NN prediction and the 'real' value to evaluate the model performance. This process is repeated 100 times to build statistics. Different inputs (P) are tested and described in Table 1. The inputs are combinations of variables that could eventually be provided to the physics-based lidar-turbulence model, i.e., $\Gamma$, $L$, and $\alpha\epsilon^{2/3}$, together with model-based computations of the lidar-turbulence measures $\sigma_{u_l}^2$, and $\sigma_{v_l}^2$.

**Table 1.** Description of the different training input datasets for the PBNNs

| Input short name | Input variables |
| --- | --- |
| P1 | $\Gamma, L, \alpha\epsilon^{2/3}, \sigma_{u_l}^2$ |
| P2 | $\Gamma, L, \sigma_{u_l}^2$ |
| P3 | $L, \alpha\epsilon^{2/3}, \sigma_{u_l}^2$ |
| P4 | $\Gamma, \alpha\epsilon^{2/3}, \sigma_{u_l}^2$ |
| P5 | $\Gamma, L, \alpha\epsilon^{2/3}$ |
| P6 | $L, \sigma_{u_l}^2, \sigma_{v_l}^2$ |
| P7 | $\sigma_{u_l}^2, \sigma_{v_l}^2$ |
| P8 | $\sigma_{u_l}^2$ |
| P9 | $L, \sigma_{u_l}^2$ |





## 3.2 The data-driven neural networks

DDNNs are constructed similarly to the PBNNs in Sect. 3.1. However, instead of using 1000 possible atmospheric states, we use the entire $m$ 10-min samples of lidar 1. The structure and type of NN are the same as that used for the PBNNs.

### 3.2.1 Cross-validation

DDNNs are also cross-validated using the same methodology as that employed for the PBNNs in Sect. 3.1.1. Here, each of datasets of measurements (one per height), each with $m$ 10-min samples, is randomly permuted and subdivided into a training and testing subsets, the NN is trained, and the generated predictions are compared with the known 'true' value, i.e., the measurement of sonic-anemometer $u$-variance that matches the height of the lidar measurement. This process is repeated 100 times. We use different inputs combining possible 10-min statistics that we can compute with this type of lidar measurements (see Table 2). Note that we also provide results (P8) for the case in which the prediction is the same as the input variable, i.e., P8 assumes that the true/sonic-equivalent $u$-variance is equal to the lidar-based $u$-variance (hereafter referred to as persistence).

**Table 2.** Description of the different training input datasets for the DDNNs

| Input short name | Input variables |
|---|---|
| P1 | $U_l, \sigma_{u_l}^2, \sigma_{v_l}^2, dU/dz, \sigma_{u_l}/dU/dz$ |
| P2 | $U_l, \sigma_{u_l}^2, \sigma_{v_l}^2$ |
| P3 | $\sigma_{u_l}^2, \sigma_{v_l}^2, dU/dz, \sigma_{u_l}/dU/dz$ |
| P4 | $\sigma_{u_l}^2, \sigma_{v_l}^2$ |
| P5 | $\sigma_{u_l}^2$ |
| P6 | $\sigma_{u_l}^2, \sigma_{v_l}^2, dU/dz$ |
| P7 | $U_l, \sigma_{u_l}^2$ |
| P8 | persistence (not a NN) |

## 4 Site and measurements

### 4.1 Site description

All measurements in this study are from instruments deployed at DTU's test station for large wind turbines in Østerild, Denmark during a 7-month period between August 2018 and March 2019. During this period, a 250-m tall meteorological mast was deployed at the south end of the row of turbine stands at the site (see Fig. 3). In addition, during the analysis period, the site consisted of seven test pads, with all turbines in a row north to south. The site is fairly flat and characterized by a mix of grasslands, urban areas, and forests with canopy heights of 10–20 m. The terrain variations around the mast are within $\pm 10$ m. The North Sea is located about 4 km north of the northernmost turbine and the waters from Limfjorden about 6 km south of the southernmost turbine.





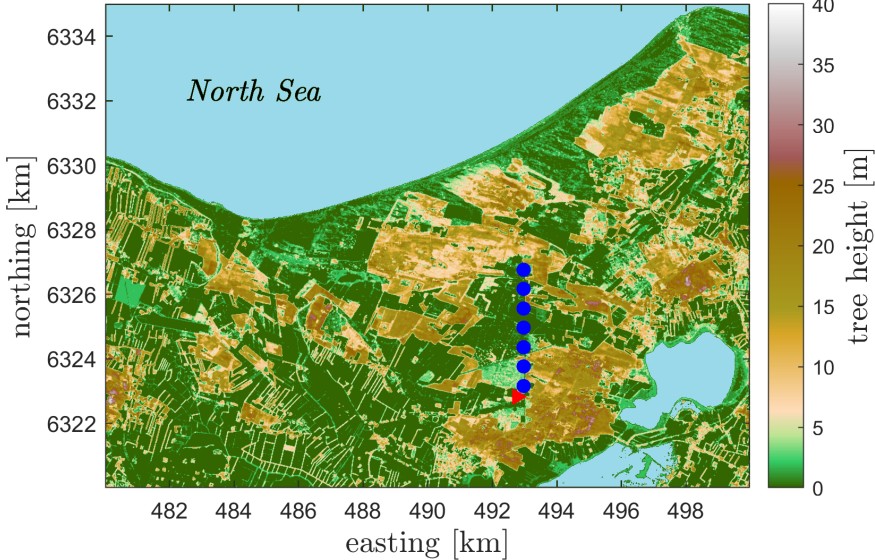

**Figure 3.** The area around the Østerild test station for large wind turbines in northern Denmark (UTM WGS84). The contour plot illustrates the tree height estimated from a digital surface model. Turbines are represented with blue circles and the mast in red triangle

## 4.2 Measurements

The mast was heavily instrumented with different sensors and here we are only interested in the sonic anemometers, which were mounted on booms at heights of 7, 37, 103, 175, and 241 m above ground level (all heights are hereafter above ground level). The mast is triangular with a side of 1.2 m. The sonic anemometers are USA-1 METEK instruments, which were side-mounted on south-oriented booms with a free boom length of 4.8 m. The sampling frequency for the sonic anemometers was

260 20 Hz. Further details about the site and the mast measurements can be found in Peña (2019).

Two ZX 300 ground-based CW lidars were installed at ≈ 11 m west of the mast. Lidar 1 was deployed at the mast during the 7-month period, whereas lidar 2 was at the mast from July 2018 to January 2019: during November 2018–January 2019, lidar 1 measured under a different configuration and data from this particular period is not used in this analysis. For the period of analysis, both lidars were configured to measure at 12 heights. The heights that match the sonic anemometers on the masts

the closest were at 39, 105, 177 and 243 m. Time synchronization was ensured during the campaign both for the mast and lidar measurements; time delays do not exceed 10 s.





# 5 Data filtering and analysis

## 5.1 Sonic anemometer measurements

The analysis is based on 10-min statistics. For the 7-month period, the 20-Hz records of the sonic anemometers are used
to obtain velocity-components' variances and covariances within each 10 min, firstly removing the instrument's in-built 2D
correction and applying the 3D correction suggested by Metek GmbH (2004). We apply yaw and pitch corrections to the time
series within each 10-min aligning the $u$-component with the mean wind. Static atmospheric stability is computed with the
Obukhov length

$$L_O = -\frac{T_s u_*^3}{\kappa g \langle w'T_s' \rangle},$$
(12)

where $T_s$ is the mean sonic anemometer temperature, $\kappa$ the von Kármán constant (0.4), $g$ the Earth's gravitational acceleration,
and the friction velocity $u_*$ is computed as

$$u_* = \left( \langle u'w' \rangle^2 + \langle v'w' \rangle^2 \right)^{1/4}.$$
(13)

We only use 10-min periods where 10000 out of the 12000 possible 20-Hz sonic records within each 10-min period are valid.

## 5.2 Lidar measurements

For both lidars, we extract the time series within each of the concurrent lidar-sonic 10-min periods. Due to the scanning speed
and the configured heights, both lidars reconstruct the horizontal wind speed and direction every ≈16 s. We remove 16-s lidar
samples if the ZX internal parameter *datavalid* is not zero or the 16-s horizontal wind speed reported by the system is above
100 m s$^{-1}$. We build the 10-min lidar-based statistics if the amount of lidar samples within the 10-min period is greater than
20 (out of a maximum value of 38). 10-min mean horizontal velocity components and directions are calculated, as well as the
variances of the velocity components, after applying yaw and pitch corrections where we align the $u$ component with the mean
wind (as we do for the sonic anemometers) using the 10-min mean direction based on the lidar. We do not use other ZX internal
parameters to further filter lidar data, such as backscatter and cloud/fog backscatter ratio, because for these two lidar datasets,
10-min mean wind speed differences between the lidars and the sonic anemometers do not relate to specific low cloud/fog
backscatter ratios. Therefore, to ensure high quality sonic-lidar datasets, for the rest of the analysis we use the 10-min lidar
statistics if at the four matching heights, the absolute difference of mean horizontal wind speed between the lidar and the sonic
anemometer is below 1 m s$^{-1}$, their mean direction absolute difference is below 20°, and the mean sonic-based direction is
within the range 60–120° or 240–300° (to avoid direct mast distortion). To filter any ambiguity in the reconstruction of the
lidar wind direction, we only use 10-min periods, in which the standard deviation of the reconstructed lidar direction is below
40°. Since we derive an estimate of the turbulence length scale based on the mean vertical wind speed gradient from the sonic
anemometers, we further restrict the analysis to 10-min periods in which the mean wind speed increases with height based on
the five sonic-anemometer heights. The amount of 10-min periods left for analysis is 3844 (out of 18743) for lidar 1 and 3519
(out of 26139) for lidar 2.



For the analysis presented in this section and in Sect. 6.1, we further classify the 10-min lidar 1 and sonic-anemometer data within three atmospheric stability conditions (stable, neutral, and unstable) based on the dimensionless stability parameter $z/L_O$ calculated from the sonic anemometer at 37 m. And within each atmospheric stability class, we further classify the data into three categories (small, medium, and large length-scale ranges) based on the approximation of the Mann turbulence length scale, i.e., $L = \sigma_u/(dU/dz)$ using the sonic-based turbulence and mean vertical wind shear measures. The different ranges used for classification into atmospheric stability and length-scale categories are presented in Table 3, together with the amount of 10-min samples for each case.

**Table 3.** Median value of the Mann turbulent length scale $L$ in meters applied in the physics-based lidar-turbulence model (Eqns. 7–10) within the atmospheric-stability and length-scale categories for the four sonic-lidar matching heights (in bold). The number of 10-min samples within each category is also given

| stability | stable | | | neutral | | | unstable | | |
|---|---|---|---|---|---|---|---|---|---|
| range | $z/L_O \geq 0.10$ | | | $-0.05 < z/L_O < 0.10$ | | | $z/L_O \leq 0.05$ | | |
| category | sta 1 | sta 2 | sta 3 | neu 1 | neu 2 | neu 3 | uns 1 | uns 2 | uns 3 |
| length scale | small | medium | large | small | medium | large | small | medium | large |
| range [m] | $L \leq 10$ | $10 < L < 16$ | $L \geq 16$ | $L \leq 23$ | $23 < L < 30$ | $L \geq 30$ | $L \leq 35$ | $35 < L < 45$ | $L \geq 45$ |
| # 10-min | 488 | 525 | 518 | 555 | 598 | 643 | 196 | 163 | 158 |
| **39** | 6.77 | 13.14 | 19.39 | 20.12 | 26.12 | 35.58 | 27.76 | 39.68 | 52.82 |
| **105** | 12.91 | 22.86 | 35.96 | 34.55 | 48.30 | 76.01 | 54.12 | 95.05 | 140.24 |
| **177** | 17.20 | 28.09 | 47.09 | 41.22 | 62.65 | 106.67 | 73.10 | 149.49 | 212.95 |
| **243** | 20.22 | 30.67 | 52.08 | 44.06 | 72.39 | 130.17 | 86.34 | 195.27 | 283.46 |

Scatter plots of 10-min means of horizontal velocity of lidar 1 and the sonic anemometers at the matching heights are shown in Fig. 4 for completeness. Figure 5 shows scatter plots of the 10-min along-velocity variance between lidar 1 and the sonic anemometer measurements at the matching heights. We use logarithmic scales since a large amount of 10-min values concentrate close to zero. As illustrated, stable atmospheric conditions generally show less variance compared to neutral and unstable atmospheric conditions at all matching heights. The highest variance values appear under neutral atmospheric conditions, where the winds are also generally the highest. Also, under stable atmospheric conditions the lidar variance measure is clearly lower than the sonic measure within the first two vertical levels in particular, whereas the lidar variance is closer to the sonic measure under unstable atmospheric conditions.

# 6 Results

## 6.1 Physics-based lidar-turbulence model

Using the physics-based lidar-turbulence model in Sect. 2.3, we compute the ratios of the lidar-to-sonic horizontal velocity variances at the four lidar-sonic matching heights. We compare these predictions with the medians of the measurements an-





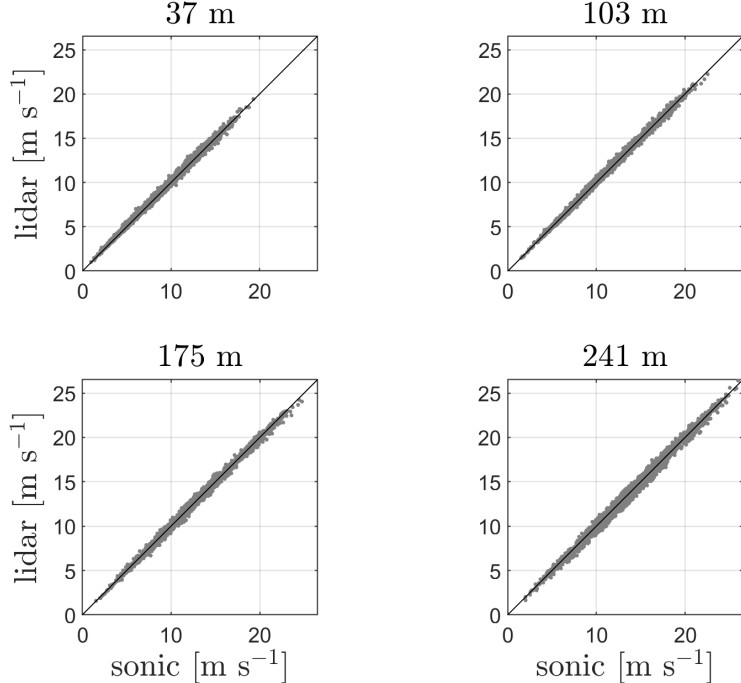

**Figure 4.** 10-min means of horizontal velocity measured by the sonic anemometers and lidar 1 for each of the matching heights

alyzed in Sect. 5 for lidar 1 within the different atmospheric-stability and length-scale categories. Based on the turbulence model, these ratios are independent of the Mann parameter $\alpha\epsilon^{2/3}$, as this only scales the turbulence level (see Eqn. 11). Since we only want to get an idea of the goodness of the physics-based lidar-turbulence model, we use the closest estimates of the $L$

values, which are the medians of the sonic-based $L$ values (see Table 3), and we assume $\Gamma = 3.0$ for all atmospheric-stability and length-scale cases, and heights.

### 6.1.1   Neutral atmospheric conditions

Figure 6 illustrates the results of the model computations and measurements for neutral atmospheric conditions and the three length-scale categories. In Fig. 6(a), which portrays the findings for the $u$-variance, the general behavior with height of lidar-

to-sonic turbulence is shown for both model results and measurements: the lidar-based turbulence is lower than the sonic-based turbulence, with the model always predicting a decrease of the lidar-to-sonic variance ratio with height. The model results are close to the observations at the first two heights, and the variance ratios from the observations do not change much between the two highest heights. At these two heights, the model results show the largest differences with the measurements. As expected, both observations and model results show larger ratios (closer to one) within the category with largest length scales (neu 3).

The comparison between model results and measurements for the case of the $v$-variance (Fig. 6b) shows similarities with respect to that of the $u$-variance. However, within the first two vertical levels, both model results and measurements clearly





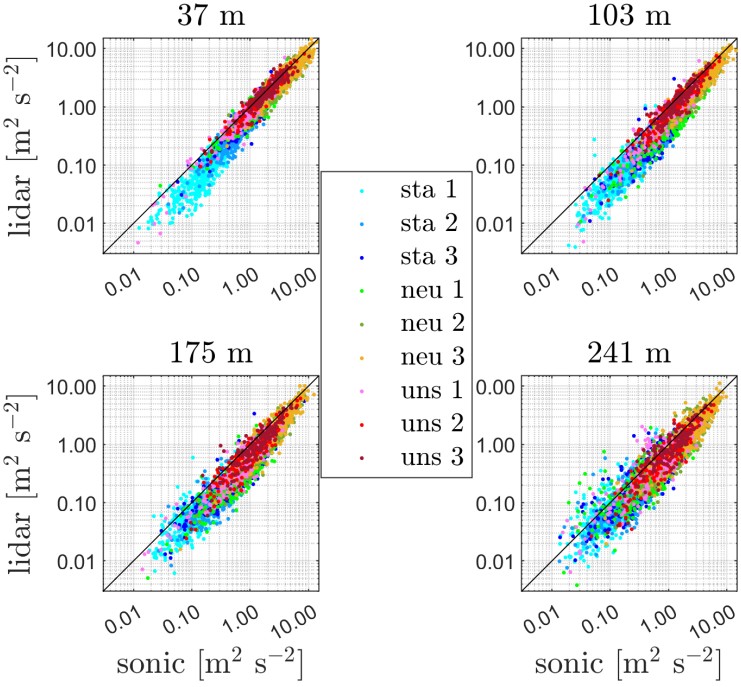

**Figure 5.** 10-min along-velocity variances measured by the sonic anemometers and lidar 1 for each of the matching heights. The different categories are color-coded and refer to those in Table 3

show larger variance ratios, which includes a case with a variance ratio larger than one that corresponds to measurements under the largest length-scale category. For these types of lidars and within the first tens of meters from the ground, the lidar-to-sonic $v$-variance ratio tends to be larger than the $u$-variance ratio (Sathe et al., 2011).

### 6.1.2 Stable atmospheric conditions

Figure 7 illustrates the results of the model computations and the measurements for stable atmospheric conditions for the three length-scale ranges. As illustrated, both model results and measurements generally show lower lidar-to-sonic variance ratios under all stable length-scale categories when compared to the results under neutral atmospheric conditions in Fig. 6. The accuracy of the model is generally worse for all stable categories compared to the neutral cases, with better model results for the cases with larger length scales compared to those with lower length-scale values. At 37 m, both measurements and model results show that within the lower length-scale range, the lidar only measures about 40% of the sonic-based velocity variances, whereas both measurements and model results are always above ≈65% at 37 m for all length-scale categories under neutral conditions.

Within stable atmospheric conditions, the differences between model results and measurements are larger at the two highest heights, particularly of the $u$-variance ratios. Also, within stable conditions at these two heights, we clearly notice a 'recovery'




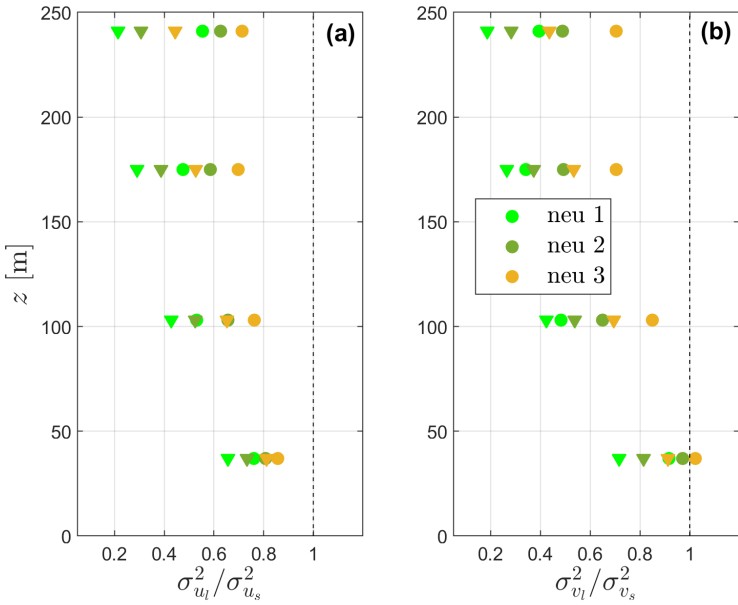

**Figure 6.** Lidar-to-sonic ratio of the $u$-variance (a) and $v$-variance (b) as function of height under neutral atmospheric conditions for the three length-scale categories (Table 3). Model results and measurements are shown in triangles and circles, respectively. Colors are in agreement with those of Fig. 5

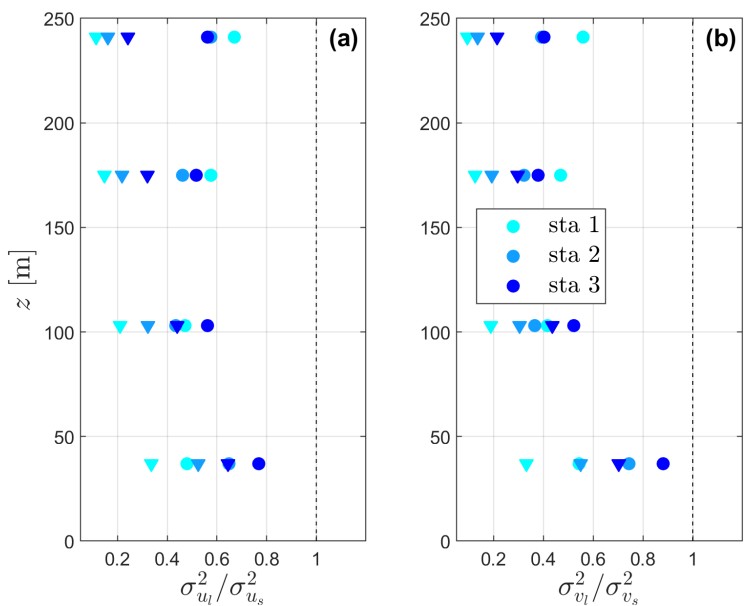

**Figure 7.** Similar to Fig. 6 but for stable atmospheric conditions





of the lidar velocity variances based on the results of the measurements, especially for the case with the smallest length scale; the model cannot mimic such recovery.

### 6.1.3 Unstable atmospheric conditions

Figure 8 illustrates the results of the model computations and the measurements for unstable atmospheric conditions for the three length-scale ranges. As illustrated, both model results and measurements generally show the highest lidar-to-sonic variance ratios under all unstable length-scale categories when compared to the results under neutral and stable atmospheric conditions in Figs. 6 and 7. The accuracy of the model under unstable conditions is also generally the highest when compared to the other two atmospheric stability classes with respect to the $u$-variance.

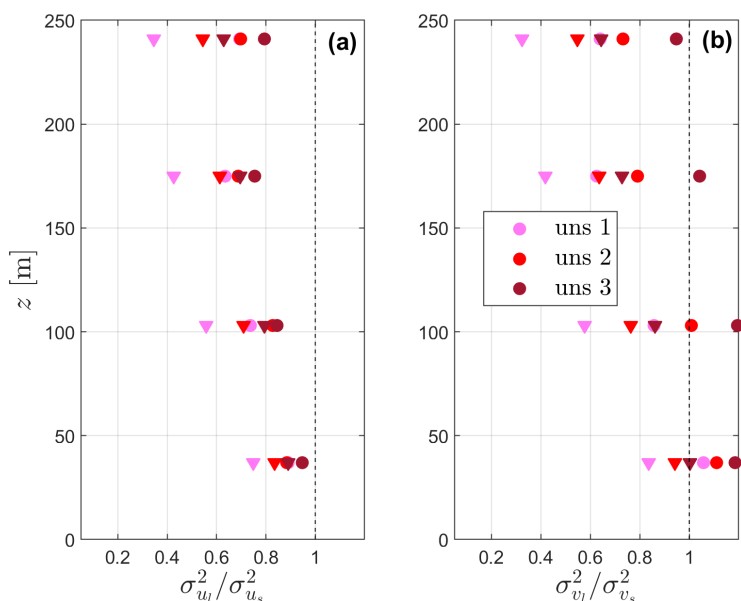

**Figure 8.** Similar to Fig. 6 but for unstable atmospheric conditions

Model computations and measurements show considerable differences for the lidar-to-sonic $v$-variance ratios. At 37 and 103
m, the lidar often measures larger $v$-variances than the sonic anemometer, up to 20% for the larger length-scale range.

### 6.2 The physics-based neural network

Four datasets of outputs from the physics-based lidar-turbulence model (Sect. 3.1) are then constructed based on the 3844 10-min statistics from lidar 1. 1000 possible atmospheric states are randomly selected from the $3844 \times 6$ possible combinations of Mann parameters that, in principle, attempt to match the turbulence climatology observed during the period of lidar 1
measurements. Figure 9 shows this random selection for the model-based datasets derived for the heights 37 and 241 m. The figure also shows the original (3844 10-min samples) sonic-based histogram of observations of $\sigma_u^2$. As illustrated, both sonic-





and model-based histograms show a much higher frequency of samples within the low-turbulence range, with decreasing samples the larger the turbulence level.

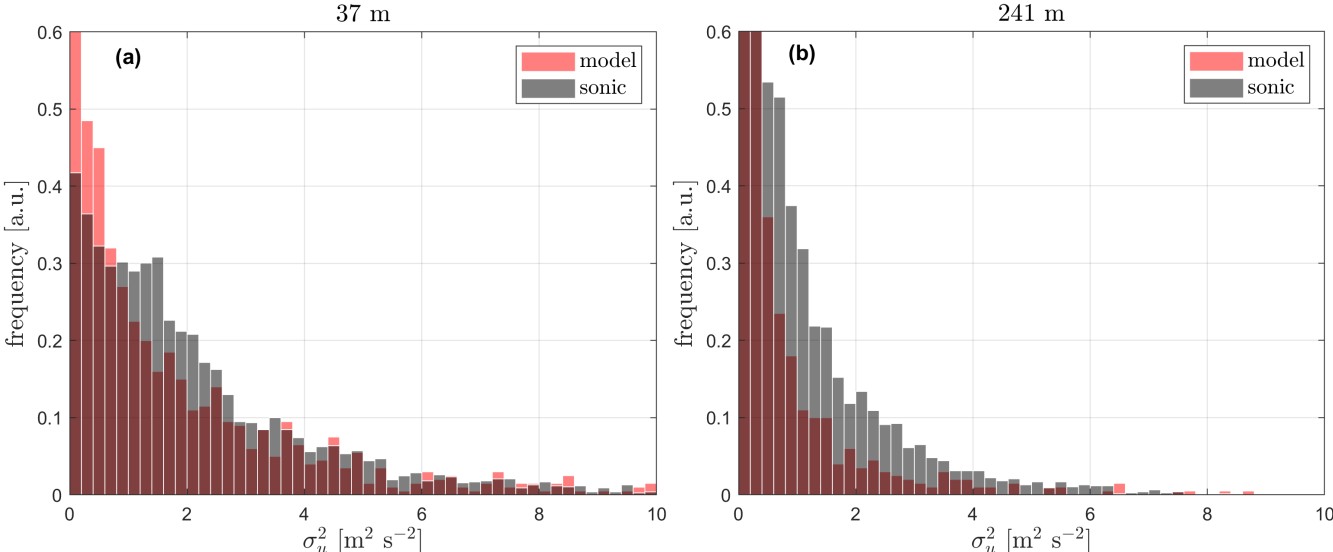

**Figure 9.** Normalized histograms of the $u$-variance from the sonic-anemometer measurements and the Mann model-based turbulence climatology at 37 (a) and 241 m (b)

### 6.2.1 Cross-validation

As described in Sect. 3.1.1, different inputs (P) can be tested for the NNs cross-validation (see Table 1). The performance results of the PBNNs for each of the tested inputs are shown as box plots in Fig. 10 for the 37-m height case. As illustrated, mape and rmse are much higher for the last three sets of inputs (P7–P9), as expected, since they use few or none of the Mann parameters for their predictions, and these are the parameters directly impacting the lidar-based turbulence prediction. P6 is an interesting case, since we could have the means to derive the three input parameters using data from lidar 1, and as shown, its performance

is rather good and similar to the first five sets of inputs (P1–P5) with median mapes far below 10% and median rmses below 0.01 m$^2$ s$^{-2}$. P1 and P5 show the lowest mapes and rmses. In principle, if we know the lidar probe-volume characteristics and the scanning height, the three inputs used by P5 are sufficient for the computation of the lidar- and sonic-based $u$-variances and so we expect the P5 performance to be the best.

### 6.2.2 Full validation

Using the same type of PBNN as that described in Sect. 3.1, we train four sets of NNs, each NN set corresponding to a combination of the three possible parameters, which we can measure or derive from the lidar 1 observations, i.e., $L$, $\sigma_{u_l}^2$, and $\sigma_{v_l}^2$ (see Table 4). Here we use the full set of 1000-samples of physics-based lidar-turbulence model inputs and outputs. The





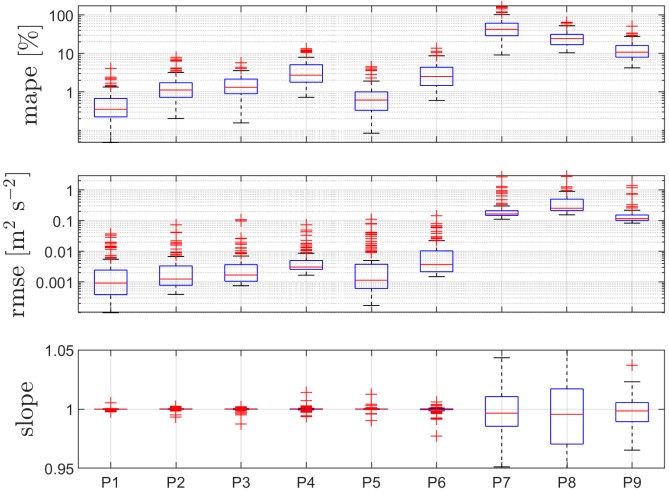

**Figure 10.** Box plots of the cross-validation for the PBNNs with different sets of inputs (see Table 1) of 100 randomly tested datasets for the height 37 m. Red solid horizontal lines represent medians, blue rectangle limits the 25th and 75th percentiles, whiskers show the maximum and minimum of the range, and red crosses represent outliers

testing is performed with the lidar 1 dataset, from which we also know the true value of the $u$-variance: the value of the sonic anemometer at the given height. The predictions are therefore carried out based on the 3844 10-min measured samples of lidar 1. We perform the analysis per measurement height. Note that the physics-based lidar-turbulence model uses the parameter $L$: we cannot accurately measure $L$ with this type of lidars, but we approximate $L$ with the expression $L_l = \sigma_{u_l}/dU/dz$. In addition, we use the $dU/dz$ values derived from the sonic-anemometer measurements, since we do not have lidar measurements below 37 m. Since the NNs are trained with initial random weights, the predictions change each iteration. The training is therefore performed 100 times over the 1000-samples input dataset and predictions are performed on the 3844 10-min measurement dataset; the result is a set of 100 3844-samples of predictions of sonic-equivalent $\sigma_u^2$ values.

**Table 4.** Description of the different training inputs for the PBNNs for the full validation

| Input short name | Input variables |
|:---:|:---:|
| P1 | $L_l, \sigma_{u_l}^2$ |
| P2 | $L_l, \sigma_{u_l}^2, \sigma_{v_l}^2$ |
| P3 | $\sigma_{u_l}^2, \sigma_{v_l}^2$ |
| P4 | $\sigma_{u_l}^2$ |
| P5 | persistence (not a NN) |

**37-m height** Results of the performance of each PBNN for the 37-m measurements are illustrated in Fig. 11. P5, i.e., persistence, does not vary, since no model is involved. It does not have poor mape or rmse (we expected this from the 1:1 comparisons



in Fig. 5), but the slope indicates a lidar-to-sonic mean bias of 16% for the $u$-variance. From the results, P4, which only uses the lidar-based $u$-variance as input, shows the lowest mape, the lowest median rmse, and the closest slope to one of all NNs.

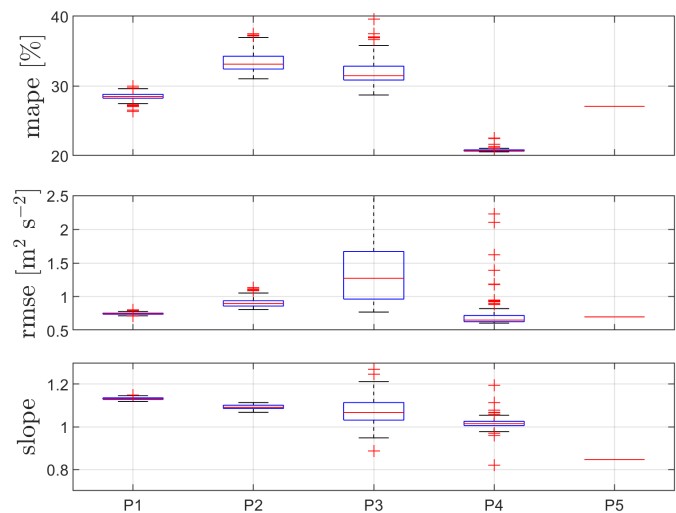

**Figure 11.** Performance of the PBNNs for different set of inputs P (see Table 4) at 37 m. Box plots of 100 trained datasets

We use the predictions from the P4-based PBNNs to construct a final prediction of lidar-based estimations of sonic-equivalent turbulence. For each of the 3844 10-min samples, we take the median of the 100 predictions of the P4-based NNs and compare the sonic-based TI values with those of the lidar and the lidar-based predictions. The results are shown in Fig. 12 for a number of mean wind speed ranges.

As illustrated, for most mean wind speed bins, the lidar predictions of TI are closer to the sonic-based TI values when compared to the original lidar measurements; this is always the case for the wind speed ranges with most measurements (3–10 m s$^{-1}$) and for most of the larger wind speed ranges ($> 10$ m s$^{-1}$). For more than half of the mean wind speed ranges, lidar-predicted TI values are slightly larger than the sonic-based TI values because the mean bias (slope) of the PBNNs is greater than one (see P4 in Fig. 11).

**103-m height** Similarly to the 37-m height case above, we show results for each of the PBNNs in Fig. 13 for the 103-m measurements. The models' performance is similar to that of the 37-m case, but with deteriorated statistics (mape, rmse, and slope). P4 is also the PBNN that performs the best and it has a median slope of 1.02, which results in a lower mean bias than persistence (0.74).

The final prediction for this height is also constructed with the P4-based PBNN. As carried out for the 37-m height, we take the median of the 100 predictions of the P4 model and compare the sonic-based TI values with those of lidar and lidar-based predictions (Fig. 14). As illustrated, for most mean wind speed bins, the lidar-based predictions of TI are also closer to the sonic-derived TI values when compared to the original lidar-based TI measurements. However, the lidar predictions match better the sonic-based values from the 5.25–5.50 m s$^{-1}$ bin onward, where most occurrences occur, whereas the lidar TI predictions highly overestimate the sonic-based TI within the lower wind speed range.



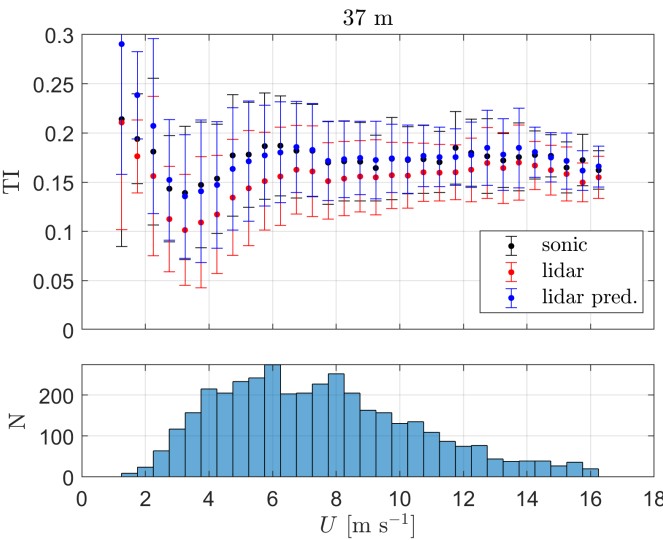

**Figure 12.** Turbulence intensity as a function of the mean wind speed from both sonic-anemometer and lidar measurements, and lidar-based predictions (lidar pred.) at 37 m. The markers show the mean and the error bars ±one standard deviation within each mean wind speed bin. N represents the number of 10-min on each mean wind speed bin

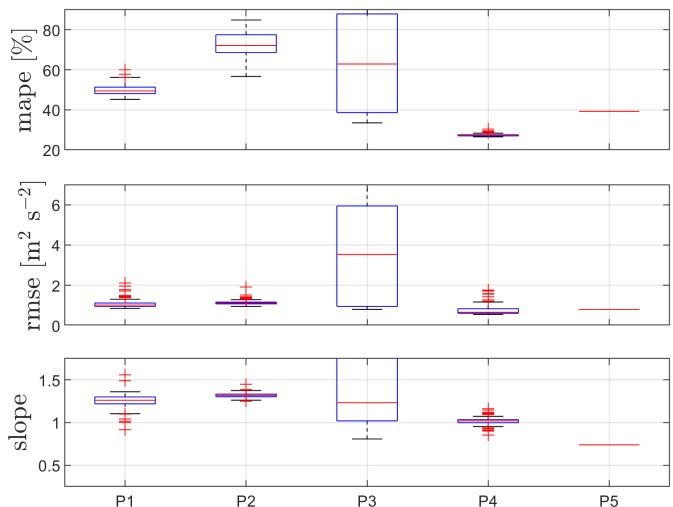

**Figure 13.** Similar to Fig. 11 but for 103 m



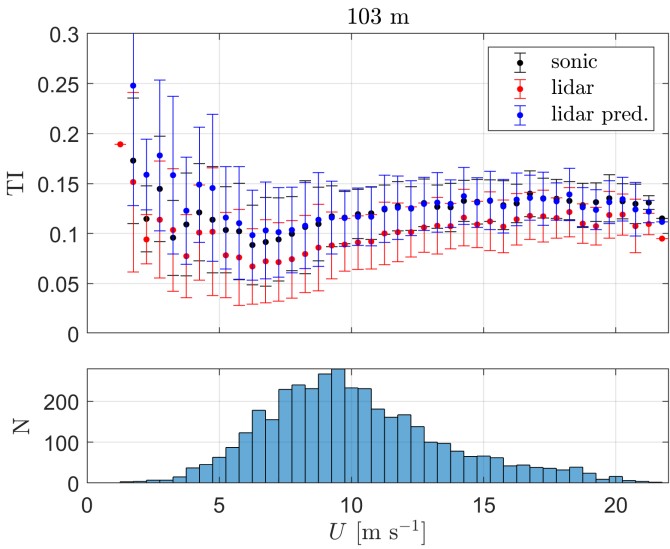

**Figure 14.** Similar to Fig. 12 but for 103 m

**175-m height** Similarly to the two previous cases, which are the two lowest measurement heights, we show results for each
of the PBNNs in Fig. 15 for the 175-m measurements. The performance of the models follows that of two lowest heights, but
with further deteriorated statistics (mape, rmse, and slope). Once again, it is P4 the PBNN that performs the best and it has a
median slope of 1.08, which results in a lower mean bias of this prediction compared to persistence (0.69).

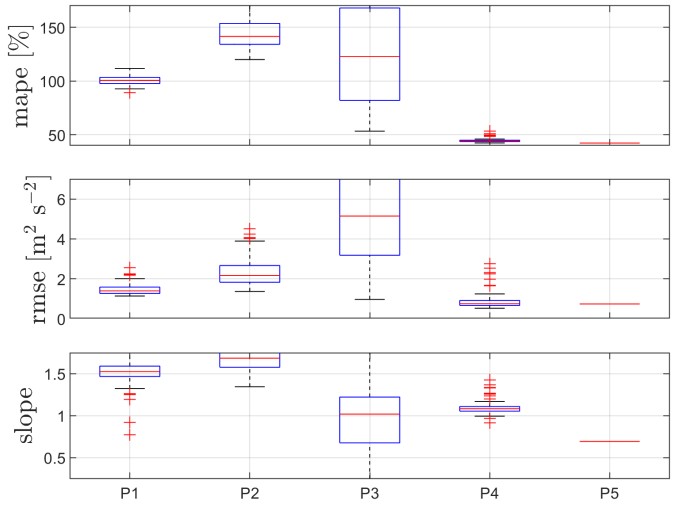

**Figure 15.** Similar to Fig. 11 but for 175 m

A final prediction is constructed using the median of the 100 predictions of the P4-based PBNN result, which we compare
against both the sonic-based and lidar-based TI values (Fig. 16). For nearly half of the mean wind speed bins (the highest half,





i.e., $U > 9.25$ m s$^{-1}$), the lidar-based TI predictions are closer to the sonic-based TI values than the original lidar-based TI measurements. The lidar-based TI predictions highly overestimate the sonic TI within the lower wind speed ranges.

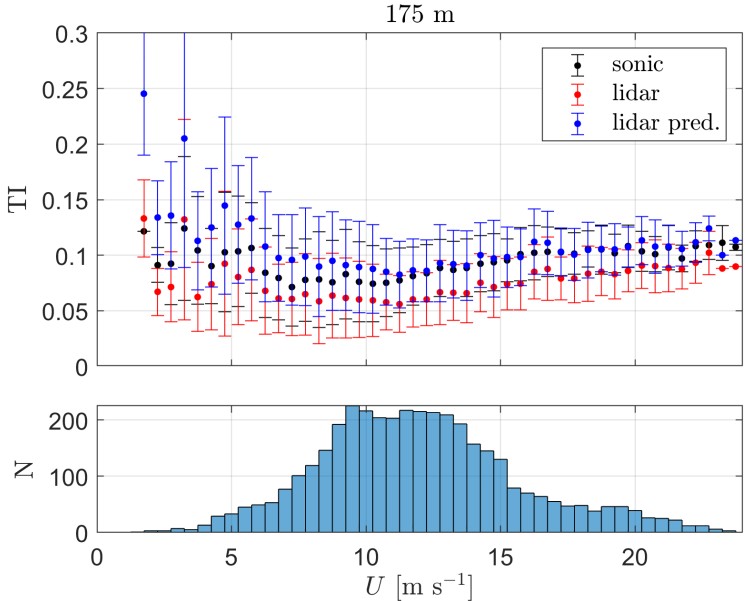

**Figure 16.** Similar to Fig. 12 but for 175 m

**241-m height** Lastly, Fig. 17 shows results for each of the PBNNs for the 241-m measurements. As expected, the models' performance follows that of the three lowest heights but with further deteriorated statistics. Once again, it is P4 the PBNN generally performing the best but with a large mape. P1's and P4's median rmses are close (about 2 m$^2$ s$^{-2}$) and P4 has a
median slope of 1.00, which results in a lower mean bias of this prediction compared to persistence (0.74).

The final prediction with the P4-based PBNN results is constructed similarly as with the previous three heights and compared to the sonic-based and lidar-based TI values (Fig. 18). At this height, it becomes clearer that the ability of the lidar-based TI predictions to match the sonic-based TI measurements is wind speed dependent as only the TI values of the highest mean wind speed ranges are those that are better predicted by the PBNN.

## 6.3  The data-driven neural network

### 6.3.1  Cross-validation

As described in Sect. 3.2.1, we test different inputs for the NNs (see Table 2). The results of the performance of the DDNNs for each of the tested inputs are shown in Fig. 19 for the 37-m height. As illustrated, persistence (P8) has the highest median mape and rmse and the lowest mean slope. Hence, assuming that the lidar-based $u$-variance is a good proxy for the sonic-equivalent
$u$-variance is not the best choice at all. Interestingly, when including the lidar-based $v$-variance (P4 vs P5), the statistics do not




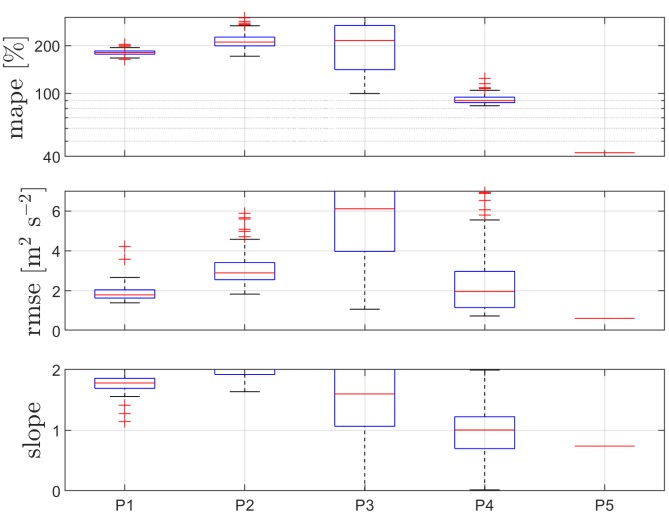

**Figure 17.** Similar to Fig. 11 but for 241 m

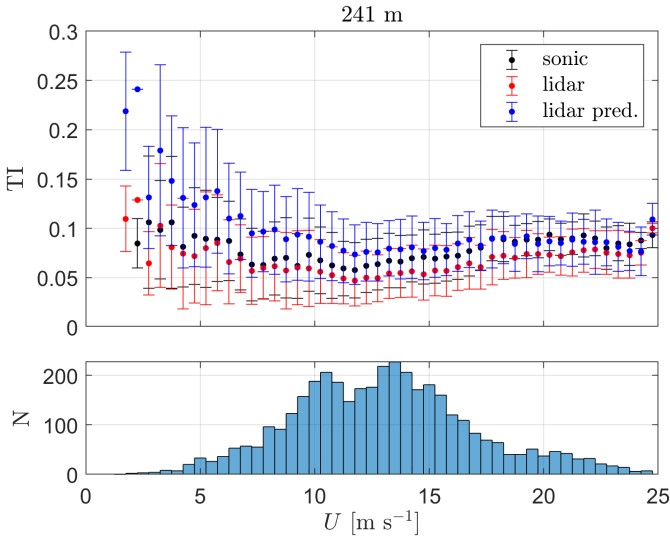

**Figure 18.** Similar to Fig. 12 but for 241 m



improve and adding the mean wind speed does not seem to help either (P5 vs P7). Also note that all the median slopes are below one, so all DNNNs predict a sonic-equivalent $u$-variance close but lower than the measured value.

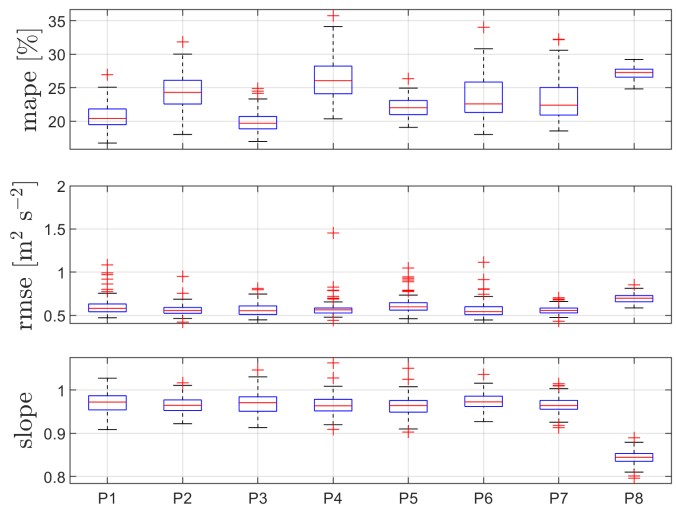

**Figure 19.** Box plots of the cross-validation for the DDNNs with different set of inputs (see Tab. 2) of 100 randomly tested datasets at 37-m height. Box plot information as in Fig. 10

### 6.3.2 Full validation

We now use the full set of 3844 10-min samples corresponding to lidar 1 measurements to construct four height-dependent NNs and evaluate them with the measurements from lidar 2 and those of the sonic anemometers. Training is performed 100 times on the 3844 10-min lidar 1 dataset and predictions are made on the 3519 10-min lidar 2 dataset; the result is a set of 100 3519 samples of predictions of sonic-equivalent $\sigma_u^2$ values.

**37-m height** Results of the performance of the DDNNs for the 37-m height using the lidar 2 measurements are illustrated in Fig. 20. In this case, P7 ($U$, $\sigma_{u_l}^2$) shows the best performance (lowest median mape and rmse, and closer to one slope). Similar but deteriorated performance is achieved by P5 ($\sigma_{u_l}^2$), which might be related to better predictions when including the mean wind speed due to the turbulence dependency on wind speed. P4, which only adds the $v$-variance to the P5 input, shows the highest mape. Using the lidar $u$-variance as proxy of the sonic-equivalent value (P8) results in nearly 10% lower $u$-variance (mean bias) compared to the sonic-based value.

For the DDNNs, we use the predictions from the P7 model to construct a final prediction of lidar-based estimations of sonic-equivalent turbulence. For each of the 3519 10-min samples, we take the median of the 100 predictions of the NNs based on P7 and compare the sonic-based TI values with those of the lidar 2 and the lidar 2-based predictions. The results are shown in Fig. 21 for a number of mean wind speed ranges.

As illustrated, for nearly all mean wind speed bins, the lidar 1-based TI predictions are in better agreement with the sonic-based TI measurements when compared to the original lidar 2-based TI measurements. For most wind speeds, the lidar predic-



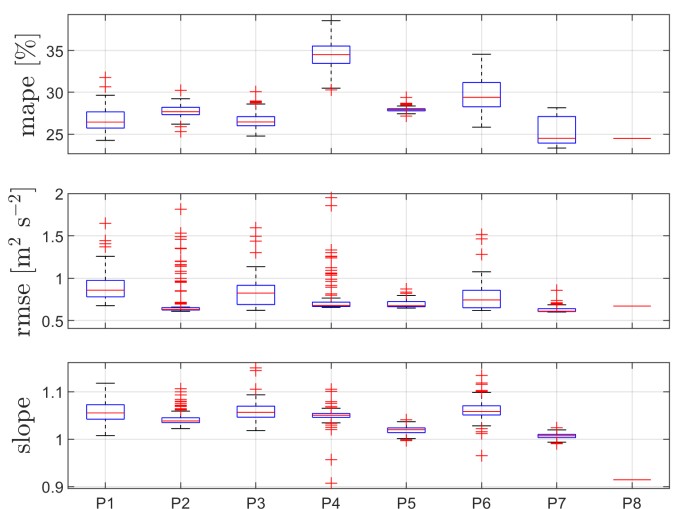

**Figure 20.** Performance of the DDNNs for different set of inputs P (see Table 2) at 37 m. Box plots of 100 trained datasets

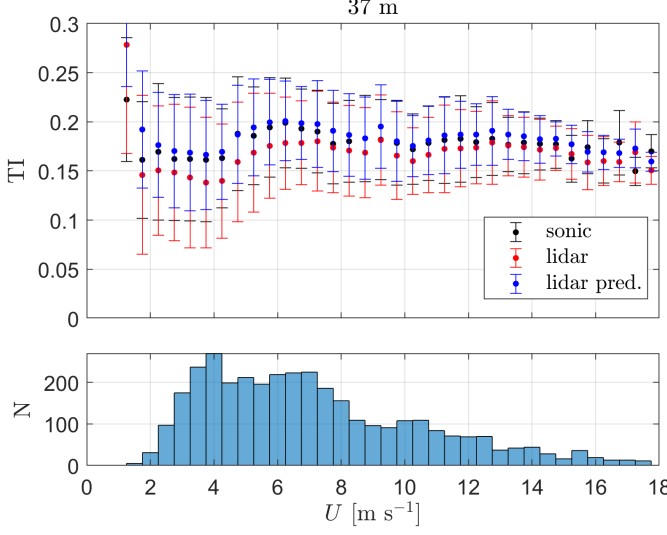

**Figure 21.** Turbulence intensity as a function of the mean wind speed from both sonic-anemometer (sonic) and lidar 2 (lidar) measurements, as well as that from lidar 2 predictions (lidar pred.) based on NNs constructed with lidar 1 measurements at 37 m. The markers show the mean and the error bars ±one standard deviation within each mean wind speed bin. N represents number of 10-min on each mean wind speed bin





tions of TI are higher than the sonic-based TI measurements, as expected, since the P7 model, as well as all other NN-based predictions, show a mean bias higher than unity (see Fig. 20).

**103-m height** Results of the performance of the DDNNs for the 103-m height using the lidar 2 measurements are illustrated in Fig. 22. Similarly to the 37-m case, P7 shows the best performance; generally, when compared to the first matching height, mapes and mean biases (slopes) are deteriorated for the NN models but P7's rmse does not seem increase. Using the lidar 2

$u$-variance as proxy of the sonic-equivalent value (P8) results in nearly 15% lower $u$-variance (mean bias) compared to the sonic-based value.

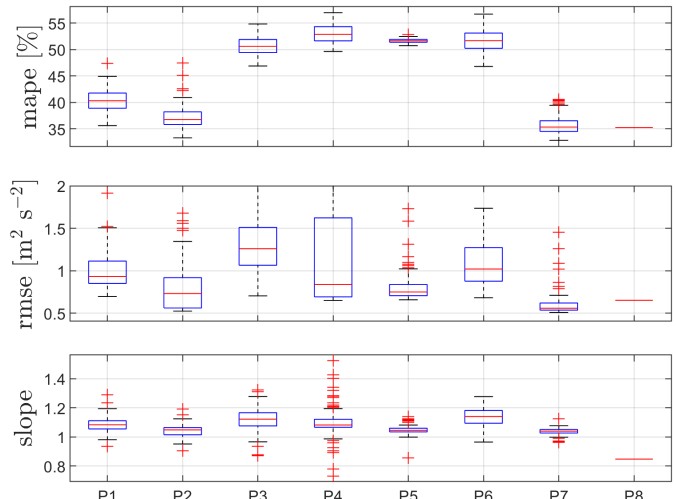

**Figure 22.** Similar to Fig. 20 but for the 103-m height

The final TI predictions for this height are also constructed with the P7-based DDNN, in a similar fashion to that performed for the 37-m height and the results shown in Fig. 23. As illustrated, for about half of the mean wind speed bins, the lidar 1-based predictions of TI are closer to the sonic-derived TI values when compared to the original lidar 2-based TI measurements.

However, the predictions do not perform better under a particular wind speed range and, under all wind speed ranges, they predict larger TI values than the original lidar 2 measurements, systematically showing larger TI predictions compared to the sonic-based TI values within nearly all mean wind speed ranges.

**175-m height** and **241-m height** We combine the results for the two highest matching vertical levels, as they show similar behavior compared to that of the lowest two matching vertical levels but with deteriorated statistics, particularly for the mape

values; however, all models' rmse values are rather similar between these two heights (see Fig. 24). For both heights, using the lidar 2 measurements as a proxy for the sonic-equivalent $u$-variance results in mean biases of about 16% and 12%, whereas the median bias of the P7 model is about 8% and 4% for the 175 and 241-m heights, respectively.

The final TI predictions for these two heights are constructed with the P7-based DDNN and the results are shown in Fig. 25. Lidar 1-based TI predictions are clearly outperformed by the original lidar 2 TI measurements; only under a few mean wind

speed bins, lidar 1-based TI predictions are closer to the sonic-based TI values. Interestingly, when comparing the TI levels



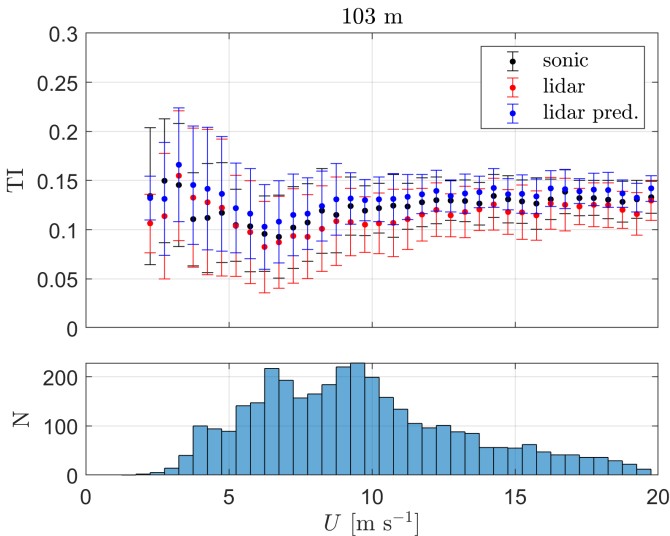

**Figure 23.** Similar to Fig. 21 but for the 103-m height

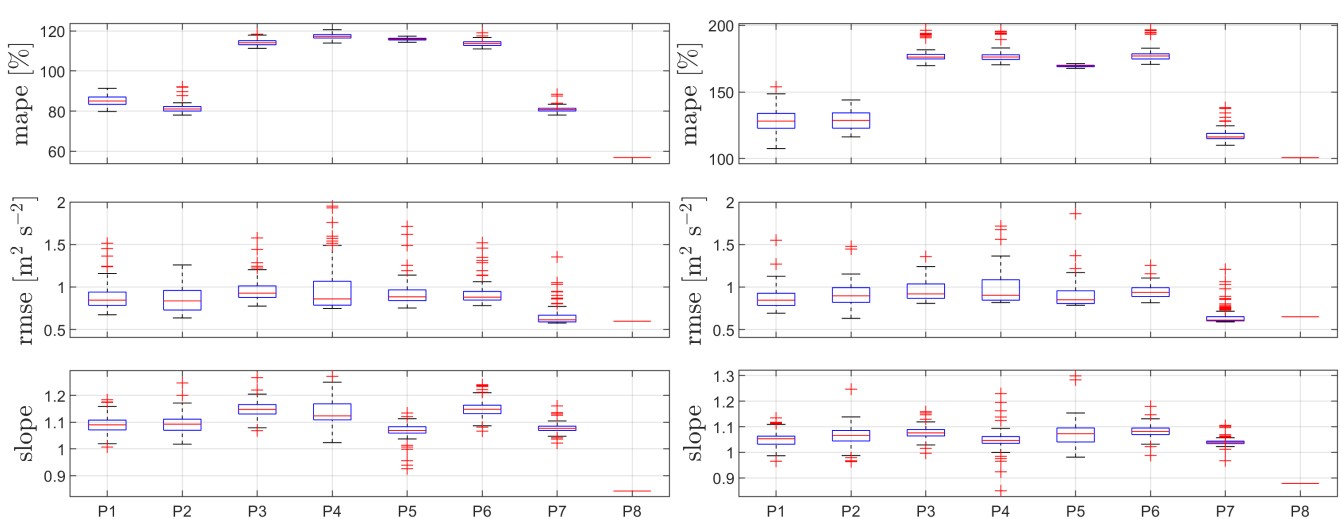

**Figure 24.** Similar to Fig. 20 but for the 175-m height (a) and 241-m height (b)




between these two heights within the low wind speed range ($U \lessapprox 5 \text{ m S}^{-1}$), the lidar-based TI levels are close or higher at 241 m. This might indicate that at least some part of the 'recovery of turbulence observed for lidar 1 is also featured by the measurements from lidar 2.

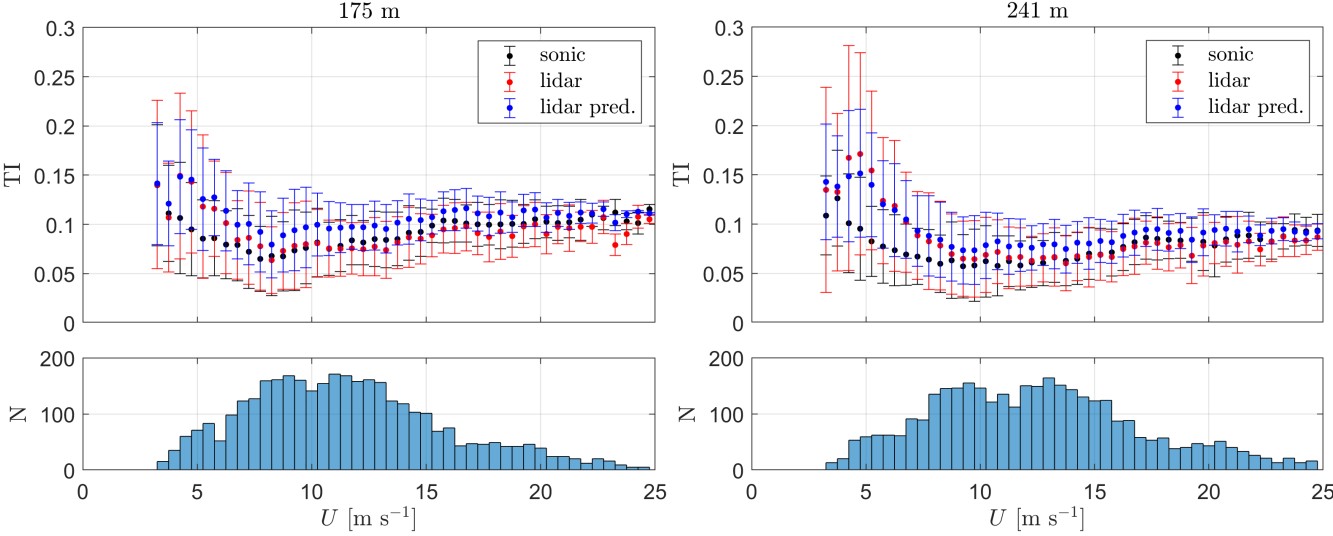

**Figure 25.** Similar to Fig. 21 but for the 175-m height (a) and 241-m height (b)

## 7 Discussion

Before discussing the results of the physics-based lidar-turbulence model, PBNNs, and DDNNs evaluated in Sect. 6, it is important to note that directly comparing the PBNNs and DDNNs here proposed is not completely fair. DDNNs are here built using information from lidar 1 only, and predictions are carried out with a fully independent new lidar dataset (in our case from lidar 2). The PBNNs use information from lidar 1 to generate the physics-based datasets that serve as input for the predictions, which ultimately are based on the full lidar 1 measurements. However, we can say that the methodology here proposed, which

involves the PBNNs, is a more practical approach than that involving the DDNNs, since we can generate new physics-based datasets to perform predictions at any site and height using site-specific lidar measurements. DDNNs are on the other hand very local as they are site- and height-dependent; however, note that we could eventually use more than a lidar or more heights to construct the NNs so eventually a lidar measurement network would benefit the DDNNs.

### 7.1 On the physics-based lidar-turbulence model

The deteriorating behavior of the physics-based lidar-turbulence model at the highest two matching heights can have different explanations. First, the results of the physics-based lidar-turbulence model depend on the goodness and characteristics of the spectral velocity tensor model used. Here, we use the Mann model, which was originally formulated to simulate the spatial





structure of stationary homogeneous turbulence under near-neutral atmospheric stability conditions and within vertical levels well inside the surface layer. Here, we are looking at atmospheric conditions beyond near-neutral stability and, in some or most cases, well above the surface layer. Further, under all atmospheric stability conditions, we are using the proxy for the Mann turbulence length scale by Kelly (2018), which is highly sensitive to estimations of the local vertical wind shear. We also use, for simplicity and to isolate the effect of the turbulent length scale, the same values of the anisotropy parameter ($\Gamma = 3.0$) at all heights, although $\Gamma$ exhibits a height dependence (Peña, 2019). Although not shown, the medians of the ratio of the $v$-to-$u$ variance based on the sonic-anemometer observations tend to be closer to unity the more unstable the atmospheric conditions are and the higher above the ground we observe. This behavior indicates that $\Gamma$ is probably lower when approaching the above mentioned conditions as turbulence becomes more isotropic. As shown in Peña et al. (2010), the three parameters of the Mann model can be determined under a wide range of turbulence conditions and heights by fitting measured sonic-anemometer velocity spectra to precomputed Mann-model spectra; here, we do not attempt this procedure, as we want to use close-to-lidar standard output to deduce the Mann parameters.

Note that the physics-based lidar-turbulence model assumes that the flow is homogeneous within the scanning volume, a condition that might not be fully valid, particularly when observing winds at the higher vertical levels at Østerild, since the scanning area of the lidar is enlarged with increasing focus distances. Also, since the lidar-to-sonic turbulence correlations deteriorate with height (see Sect. 5), the lidar Doppler radial velocity spectrum might be more sensitive to noise impacting mostly the lower end of radial velocity bins. Such impact artificially increases the radial and reconstructed velocity variances, which could explain the recovery of turbulence at the highest measurement levels. Finally, 2D turbulence might be present, particularly when measuring above 100 m; the Mann model, which is a 3D turbulence model, could be missing some part of the velocity variability at low-frequencies (Cheynet et al., 2018).

### 7.2 On the physics-based neural networks

The abilities of the PBNNs depend on both the accuracy of the physics-based lidar-turbulence model and on the accuracy/quality of the lidar turbulence measurements. Here we quality-filter lidar measurements by using the 10-min samples in which the difference in mean wind speed between lidar 1 and the sonic anemometers is below a threshold. The ratios of lidar-to-sonic $u$- and $v$-variance based on measurements show a recovery, particularly above 103 m, which the model cannot predict. The recovery can also be seen clearly in the TI predictions. Within the low mean wind speed range, the original lidar TI is consistently lower than the sonic TI at 37 m; the difference in TI between these two estimates decreases with increasing height. The consequence is that the lidar-based prediction overcorrects the sonic-based TI value because the physics-based lidar-turbulence model, on the contrary to the measurements, cannot predict turbulence recovery at these heights.

### 7.3 On the data-driven neural networks

The abilities of the DDNNs depend on both the accuracy/quality of the lidar turbulence measurements used to construct the NNs (in this case lidar 1) and that of the lidar turbulence measurements used for predictions (in this case lidar 2). Note that the differences between the lidar 2 and sonic-based TI values (Figs. 21–25) are smaller than those between the lidar 1 and sonic-





based TI values (Figs. 12–18). These TI differences also decrease with height, suggesting that the turbulence recovery observed in lidar 1 measurements might be even stronger for lidar 2 measurements; apart from a couple of mean wind speed bins, the DDNN-based lidar TI predictions are always higher than the original lidar TI measurements. For lidar 2 measurements, we apply the same filtering techniques as for lidar 1 measurements; if noise contamination of the Doppler radial velocity spectrum

is present, this seems to more strongly impact the radial velocity fluctuations of lidar 2.

## 8  Conclusions

The physics-based lidar-turbulence model agrees well with the lidar-1 and sonic-anemometer measurements under a number of atmospheric stability conditions and length-scale ranges within the two first lidar-sonic matching heights, although we do not try to adjust the spectral model parameters, which are the basis of the model. Depending on the length-scale range and

atmospheric stability conditions, the level of agreement (i.e., mean bias) deteriorates particularly at the highest two matching heights.

The physics-based neural networks, which are trained by combining the lidar 1 measurements with the physics-based lidar-turbulence model and applied to correct the lidar 1 measurements, accurately predict the TI levels measured by the sonic anemometers at the two first matching heights within the broad range of mean wind speeds. With increasing vertical level, the

predictions are outperformed by the TI estimates based on the uncorrected lidar 1 measurements within the lowest mean wind speed range.

The data-driven neural networks, trained with lidar 1 measurements and applied to correct the lidar 2 measurements, accurately predict the TI levels measured by the sonic anemometers at the two first matching heights within most mean wind speed bins. At the two highest matching heights, the uncorrected lidar 2 TI measurements are generally closer to the sonic-based

values than the lidar 1-based predictions, which overestimate the standard turbulence measures.

Although the results at the two highest matching heights are not encouraging for the DDNNs in particular, both presented methodologies appear as possible countermeasures of the lidar-turbulence paradox and can be further 1) improved, e.g., by better matching the local turbulence conditions at which predictions are performed, enlarging the datasets used for training, or improving the neural network architecture itself, and 2) extended, e.g., by using a lidar network for training the neural

networks. The main objective of the study is to demonstrate that these approaches can be used to correct lidar-based turbulence measures.

*Data availability.* Mast and lidar measurements are proprietary data. The four datasets that include the theoretical lidar-based and sonic-based velocity variances for each of the lidar-sonic matching heights and the combination of Mann turbulence parameters are available at https://figshare.com/s/0b82ac46e49215b81bd2.



*Author contributions.* AP developed the methods, performed the analysis, and wrote the initial draft. GY and VM quality-controlled the lidar and mast measurements, provided the datasets used for the analysis, conceptualized the project, and acquired the funding and resources for the work. All authors revised and edited the manuscript.

*Competing interests.* Alfredo Peña is an associate editor of the Wind Energy Science journal.

*Acknowledgements.* We would like to acknowledge Energinet and Ørsted for providing us with the resources to pursue this work.



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
