# Peer review of "On the lidar-turbulence paradox and possible countermeasures"

_Wind Energy Science, 2024_

## Referee Comment (RC1)

**Review of the manuscript wes-2014-108 entitled "On the lidar-turbulence paradox and possible countermeasures" by A. Peña, G. G. Yankova, and V. Mallini**

This article addresses LiDAR-turbulence paradox, i.e. the underestimation of the LiDAR-based turbulence intensity due to the probe volume averaging. Based on the data collected during a months-long campaign by two continuous-wave profiling LiDARs and four sonic anemometers, the authors adopt a neural network (NN) algorithm to correct the LiDAR's along-velocity variance. The problem is addressed at three different levels: first, utilizing a physics-based turbulence model to efficiently reproduce the LiDAR-to-sonic turbulence ratio; second, the physics-based model is used to train a physics-based NN (PBNN) on LiDAR turbulence data to correct the probe-volume averaging effect and positively assess them against the sonic anemometry; finally, the data from one LiDAR are used to train a data-driven NN (DDNN) to correct the turbulence prediction from a second co-located LiDAR and compare the latter against sonic anemometry. The article is well-written, the results are encouraging, and the limitations of the current analysis are thoroughly discussed. Thus, I recommend this article for publication after few major revisions I have detailed in the remainder of this review.

**General comments**

- Eq. 1: If you assume a right-handed reference frame (i.e. with the $z$-axis pointing upward), the radial wind speed is: $v_r = u \cos \phi + w \sin \phi$. Please cross check this equation.

- Eq. 3 (Line 115): Should the argument of the complex exponential have a negative sign?

- Line 124: I would explicitly write the Mann's model of the spectral tensor.

- Lines 133-134: Which values of $L$ and dissipation rate did you use?

- Line 164: To improve the overall clarity of the manuscript, I suggest moving the "Methods" section after the site description section, since the content of Section 4 is mentioned several times in Section 3.

- Line 197: Please provide a reference for the estimation of the dissipation rate.

- Line 261: I believe a schematic of the experiment (or pictures) is useful to the reader to better understand the remainder of this study.

- Line 265: A maximum time lag of 10 s is acceptable when second-order statistics are cross-validated among different instruments. However, I would not mention "time synchronization" for this study as, to my understanding, the different instruments were not sampling based on a common GPS time stamp.

- Line 306: The agreement between mean velocity measured by sonic and Lidar looks excellent. Please add the slope, intercept and coefficient of determination to quantify the agreement.

- Lines 308-312: In Fig. 4, the information may be better stated quantifying the linear regression parameters between lidar- and sonic-based variance. You can just provide one global value for each height (regardless of the stability and length scale), or specific values

for each data subset. I leave this up to the authors' decision, but I believe that the initial comparison between sonic and lidar variance estimates requires a thorough quantification.

- Line 347: Previously (Line 320) you mentioned that a common value of $\Gamma = 3$ is used across all stability and length scale cases. Considering that the thermal stability influences the isotropy/anisotropy of the energy-carrying turbulent structures, would you observe any difference in Fig. 7 letting $\Gamma = 2$ or $\Gamma = 4$?

- Lines 388-389: Do you have an explanation of why the case with the lowest number of inputs (P4) returns the best agreement? During the cross-validation of the PBNN the result was opposite (case P8 in Table 1).

- Lines 394-398: In Fig. 12 (and following) I would rather just plot the error bars of the along-velocity variance against the wind speed, instead of the TI (which is not independent from the mean wind speed). The error on the TI inevitably carries the error on the mean velocity (tough small), whereas here you want to highlight only the error on the second-order turbulence statistic.

- Line 406: I agree that having a large number of samples within the $5.25 - 5.50$ m s$^{-1}$ bin improves the agreement with the sonic-based values. However, I see that you have a good agreement also for the highest bins, where the number of occurrences is much smaller. Thus, I don't think that the occurrences are the only reason why the agreement is poorer at low wind speed. My suggestion is to separate the along-velocity variance from the wind speed and see how the results change (see my previous comment).

**Technical comments**

- Table 3: The header of the last column should report $\frac{z}{L_O} \leq -0.05$.

- Figure 9: I do not understand the unit on the ordinate axis. The figure's caption states that the histogram is normalized, but the sum of all the bar values is clearly greater than 1 (or 100%). Also, please change the upper limit of the $y$-axis to show the exact values of the left-most bar.

- Line 374: I found this sub-subsection a little bit dispersive in terms of results visualization. If the authors think it is possible, I suggest concentrating the top panels of Fig. 12, 14, 16 and 18 in a single figure, the bottom panels of these figures in another one and do the same for Figs. 11, 13, 15 and 17. The current back-and-forth between the text and the figures makes this portion of the manuscript a little bit hard to follow.

- Line 410: Please replace "that of two" with "the ones of the two [...]".

- Line 471: Please correct the equation: $U \leq 5$ m s$^{-1}$.

- Line 526: Before summarizing the results of this study, it is useful to start the Conclusion with a summary of the experimental setup and data analysis methods used in this work.

---

## Referee Comment (RC2)

**Review Report: Wind energy science**

*On the lidar-turbulence paradox and possible countermeasures*

This manuscript explores valuable methods for correcting CW Doppler lidar turbulence measurements through both a physics-based model and neural networks, utilizing data from CW lidars and sonic anemometers across various heights. While the lidar-turbulence model demonstrates good performance at lower altitudes, it faces challenges at higher levels, attributed to turbulence filtering and contamination effects. The neural networks appear to enhance the accuracy of lidar-based turbulence measurements, although some biases persist, particularly at elevated wind speeds.

Overall, this paper provides a contribution to the field by addressing the lidar-turbulence paradox with innovative solutions and providing a comprehensive analysis of lidar performance across different altitudes. It builds on existing literature while introducing new methodologies that could pave the way for more accurate and reliable lidar-based turbulence measurements. Several detailed comments are provided below.

**Comments**

1.      L1-2 – "*lidar-based…standard turbulence measures*". I understand that it is impossible to provide accurate definitions in an abstract; however, this can be misleading because turbulence measurements can be performed with different instruments with different temporal and spatial resolutions, e.g. from lidars down to micro hot-wire anemometers; nonetheless, they are still considered turbulence measurements. I would rephrase it as "…that corrects lidar turbulence measurements to enable adequate turbulent statistics for atmospheric and wind energy applications", or something similar.

2.      L50 – I would add $\langle u'w' \rangle$ are generally negative (Chowdhuri, S. and Deb Burman, P.K., 2020. Representation of the Reynolds stress tensor through quadrant analysis for a near-neutral atmospheric surface layer flow. *Environmental Fluid Mechanics*, *20*(1), pp.51-75.; Shig, L., Babin, V., Shnapp, R., Fattal, E., Liberzon, A. and Bohbot-Raviv, Y., 2023. Quadrant analysis of the Reynolds shear stress in a two-height canopy. *Flow, Turbulence and Combustion*, *111*(1), pp.35-57.)

3.      L53 – If you define $v_r = u\cos\phi + w\sin\phi$, then you should get $\sigma_{v_r} = \sigma_u^2\left(\cos^2\phi + \frac{1}{4}\sin^2\phi - \frac{1}{6}\sin 2\phi\right)$. If that's the case, you should revise the following discussion with different $\phi$. Please cross-check.

4.      L64 – "…eddies with most energy". Actually, the energy-containing eddies are those at larger scales, larger than those belonging to the inertial subrange. I would rather say…a probe volume small enough to probe turbulence processes at small scales, ideally down to those responsible for dissipation, which are proportional to the Kolmogorov scale.

5.      L197 – Can you provide references about the relationship between turbulence intensity and height? How much does it differ from other theories, e.g. Townsend wall-attached eddy hypothesis where $\sigma_u^2 = B_1 - A_1\log\frac{z}{\delta}$?

6.      L 317 – In Fig. 4 – make sure y-axes and x-axes have the same scale.

7.      Conclusions: You should provide an overall summary of work describing the strategy, the objectives, not only the results. Please rework on the conclusions.

---

## Author Comment (AC1)

**Response to the comments from referee 1**

Thanks for your very positive comments on our manuscript. Here our response to each of your comments. The response is given in blue color.

Best regards,
The authors
* * *
This article addresses LiDAR-turbulence paradox, i.e. the underestimation of the LiDAR-based turbulence intensity due to the probe volume averaging. Based on the data collected during a months-long campaign by two continuous-wave profiling LiDARs and four sonic anemometers, the authors adopt a neural network (NN) algorithm to correct the LiDAR's along-velocity variance. The problem is addressed at three different levels: first, utilizing a physics-based turbulence model to efficiently reproduce the LiDAR-to-sonic turbulence ratio; second, the physics-based model is used to train a physics-based NN (PBNN) on LiDAR turbulence data to correct the probe-volume averaging effect and positively assess them against the sonic anemometry; finally, the data from one LiDAR are used to train a data-driven NN (DDNN) to correct the turbulence prediction from a second co-located LiDAR and compare the latter against sonic anemometry. The article is well written, the results are encouraging, and the limitations of the current analysis are thoroughly discussed. Thus, I recommend this article for publication after few major revisions I have detailed in the remainder of this review.

**General comments**

- Eq. 1: If you assume a right-handed reference frame (i.e. with the $z$-axis pointing upward), the radial wind speed is: $v_r = u \cos \phi + w \sin \phi$. Please cross check this equation.

The Equation in line 47 of the original submission is correct for an upwind lidar beam (as the one we show in the figure). The equation mentioned by the reviewer is correct for the downwind case or for the case with 'negative' $\phi$ angles as we mentioned in the original submission in lines 54–55. To clarify better this, we now add "i.e., pointing upwind," just before the above equation is mentioned and add "or when the lidar points downwind" after "For the cases with negative $\phi$ values" later in the paragraph.

- Eq. 3 (Line 115): Should the argument of the complex exponential have a negative sign?

Yes, it is now corrected as suggested. Thanks for noticing.

- Line 124: I would explicitly write the Mann's model of the spectral tensor.

We do not think that writing fully the elements of the Mann tensor will aid to the manuscript. It is explicitly written in Mann [1994], which we refer to in the same lines.

- Lines 133-134: Which values of $L$ and dissipation rate did you use?

As suggested, we added the values of $L$ used for the computations. The ratio of the variances is independent on the dissipation parameter value for the Mann model. This is also now stated along the same lines.

- Line 164: To improve the overall clarity of the manuscript, I suggest moving the "Methods" section after the site description section, since the content of Section 4 is mentioned several times in Section 3.

As suggested, we moved the Methods section after the Data filtering and analysis section.

- Line 197: Please provide a reference for the estimation of the dissipation rate.

As suggested, we now add the text '... which is based on the form $\epsilon \kappa z / u_*^3 = 1$ from similarity theory [Stull, 1988]' along those lines

- Line 261: I believe a schematic of the experiment (or pictures) is useful to the reader to better understand the remainder of this study.

These are ground-based commercial profiling lidars besides a tall meteorological mast; for this particular 'standard' scanning mode and standard procedure of evaluation, we do not think that a figure will aid to the manuscript.

- Line 265: A maximum time lag of 10 s is acceptable when second-order statistics are cross-validated among different instruments. However, I would not mention "time synchronization" for this study as, to my understanding, the different instruments were not sampling based on a common GPS time stamp.

As mentioned by the reviewer, there is not a lidar/mast common GPS time so we have deleted the time synchronization part.

- Line 306: The agreement between mean velocity measured by sonic and Lidar looks excellent. Please add the slope, intercept and coefficient of determination to quantify the agreement.

It looks excellent because we only use data where the difference between lidar and sonic mean wind speed is less than 1 m $s^{-1}$ as we stated in lines 290–291 of the original submission. We show the comparison/data for completeness only and find that it will be unfair to evaluate the agreement when we have on purpose biased the comparison.

- Lines 308-312: In Fig. 4, the information may be better stated quantifying the linear regression parameters between lidar- and sonic-based variance. You can just provide one global value for each height (regardless of the stability and length scale), or specific values for each data subset. I leave this up to the authors' decision, but I believe that the initial comparison between sonic and lidar variance estimates requires a thorough quantification.

We understand the comment of the reviewer but we believe such quantification does not provide insights about the lidar-turbulence problem itself. For this type of scanning and lidar, sonic-lidar velocity variance comparisons, contrary to mean speed comparisons, should be biased and should have scatter; values that quantify the bias and scatter are function of the amount of contamination and filtering, which is dependent of the turbulence characteristics (the latter dependent on the length scale and anisotropy on the view of the Mann model). Low correlations in these comparisons could simply mean that the range of turbulence conditions explored is broad, and not particularly that the lidar is not measuring as it should.

- Line 347: Previously (Line 320) you mentioned that a common value of $\Gamma = 3$ is used across all stability and length scale cases. Considering that the thermal stability influences the isotropy/anisotropy of the energy-carrying turbulent structures, would you observe any difference in Fig. 7 letting $\Gamma = 2$ or $\Gamma = 4$?

Yes, as already hinted when presenting Fig. 2, there is a dependency on $\Gamma$, not only in Fig. 7 but on Figs. 6 and 8 of the original submission. We only use a fixed $\Gamma$ value for these three plots to show how the velocity variance ratios change with height and length scale. However, we vary $\Gamma$ when creating the datasets based on the physics-based model for the NNs.

- Lines 388-389: Do you have an explanation of why the case with the lowest number of inputs (P4) returns the best agreement? During the cross-validation of the PBNN the result was opposite (case P8 in Table 1).

This is because during the cross-validation (Sect. 3.1.1 of the original submission), we only use model-based datasets; however for the full evaluation we use lidar observations to estimate the parameters that the physics-based model needs or outputs.

- Lines 394-398: In Fig. 12 (and following) I would rather just plot the error bars of the along-velocity variance against the wind speed, instead of the TI (which is not independent from the mean wind speed). The error on the TI inevitably carries the error on the mean velocity (tough small), whereas here you want to highlight only the error on the second-order turbulence statistic.

We agree with the reviewer that the parameter of study should be the along-velocity variance and not the TI. However, TI is the standard turbulence measure of the community (for good or wrong reasons) and we believe the results have much more impact showing this metric (also TI vs $U$ is a metric for turbine site selection). Since we make sure that the lidar exhibits an excellent agreement with the sonic data for the mean wind speed, the TI deviations are, in our particular case, driven by the variance deviations mostly.

- Line 406: I agree that having a large number of samples within the 5.25–5.50 m/s bin improves the agreement with the sonic-based values. However, I see that you have a good agreement also for the highest bins, where

the number of occurrences is much smaller. Thus, I don't think that the occurrences are the only reason why the agreement is poorer at low wind speed. My suggestion is to separate the along-velocity variance from the wind speed and see how the results change (see my previous comment).

The reviewer is right that this is not the only reason. In the mentioned lines, we only state the observation: high deviations occur within this range of wind speeds, which show low number of samples; later in the Discussion section, we provide some insights to other issues/challenges that might impact the results on the lidar TI predictions.

**Technical comments**

- Table 3: The header of the last column should report $z/L \leq -0.05$.

Changed as suggested.

- Figure 9: I do not understand the unit on the ordinate axis. The figure's caption states that the histogram is normalized, but the sum of all the bar values is clearly greater than 1 (or 100%). Also, please change the upper limit of the y-axis to show the exact values of the left-most bar.

We now add that a.u. refers to arbitrary units in the caption. We limited the $y$-axis to 0.6 to clearly show the difference in the modeled and observed variance histogram, as the frequency of the first bin is nearly double that of the second bin, and to ease the comparison between the two heights. We also add in the caption that normalization is performed on each bin by dividing the number of elements in the bin by the product of the total number of inputs and the bin width. In this way one can compare the relative frequency of each bin from two datasets with different samples.

- Line 374: I found this sub-subsection a little bit dispersive in terms of results visualization. If the authors think it is possible, I suggest concentrating the top panels of Fig. 12, 14, 16 and 18 in a single figure, the bottom panels of these figures in another one and do the same for Figs. 11, 13, 15 and 17. The current back-and-forth between the text and the figures makes this portion of the manuscript a little bit hard to follow.

We understand the comment of the reviewer. The final layout of the WES papers is in a two-column format so the papers in review do not reflect the final layout. If accepted, we will try to make sure (as much as possible by commenting to the technical editors during the proof-reading stage) that figures can be found after the text describing them within the same page.

- Line 410: Please replace "that of two" with "the ones of the two [...]".

As suggested, we replace the text but we use "the one of the two" instead.

- Line 471: Please correct the equation: $U \leq 5$ m s$^{-1}$.

Corrected as suggested.

- Line 526: Before summarizing the results of this study, it is useful to start the Conclusion with a summary of the experimental setup and data analysis methods used in this work.

As suggested, we now add a summary of the work and the ideas before presenting the results

**References**

J. Mann. The spatial structure of neutral atmospheric surface-layer turbulence. *J. Fluid Mech.*, 273:141–168, 1994.

R. B. Stull. *An introduction to boundary layer Meteorology*. Kluwer Academic Publishers, 1988.

---

## Author Comment (AC2)

**Response to the comments from referee 2**

Thanks for your very positive comments on our manuscript. Here our response to each of your comments. The response is given in blue color.

Best regards,
The authors
* * *
This manuscript explores valuable methods for correcting CW Doppler lidar turbulence measurements through both a physics-based model and neural networks, utilizing data from CW lidars and sonic anemometers across various heights. While the lidar-turbulence model demonstrates good performance at lower altitudes, it faces challenges at higher levels, attributed to turbulence filtering and contamination effects. The neural networks appear to enhance the accuracy of lidar-based turbulence measurements, although some biases persist, particularly at elevated wind speeds. Overall, this paper provides a contribution to the field by addressing the lidar-turbulence paradox with innovative solutions and providing a comprehensive analysis of lidar performance across different altitudes. It builds on existing literature while introducing new methodologies that could pave the way for more accurate and reliable lidar-based turbulence measurements. Several detailed comments are provided below.

**Comments**

1. L1-2 – "lidar-based...standard turbulence measures". I understand that it is impossible to provide accurate definitions in an abstract; however, this can be misleading because turbulence measurements can be performed with different instruments with different temporal and spatial resolutions, e.g. from lidars down to micro hot-wire anemometers; nonetheless, they are still considered turbulence measurements. I would rephrase it as "...that corrects lidar turbulence measurements to enable adequate turbulent statistics for atmospheric and wind energy applications", or something similar.

We now rephrase the sentence nearly as suggested.

2. L50 – I would add $\langle u'w' \rangle$ are generally negative (Chowdhuri, S. and Deb Burman, P.K., 2020. Representation of the Reynolds stress tensor through quadrant analysis for a near-neutral atmospheric surface layer flow. Environmental Fluid Mechanics, 20(1), pp.51-75.; Shig, L., Babin, V., Shnapp, R., Fattal, E., Liberzon, A. and Bohbot-Raviv, Y., 2023. Quadrant analysis of the Reynolds shear stress in a two-height canopy. Flow, Turbulence and Combustion, 111(1), pp.35- 57.)

We now add one of the references suggested by the reviewer.

3. L53 – If you define $v_r = u \cos\phi + w \sin\phi$, then you should get $\sigma_{v_r} = \sigma_u^2 \left( \cos\phi^2 + 1/4 \sin\phi^2 - 1/6 \sin 2\phi \right)$. If that's the case, you should revise the following discussion with different $\phi$. Please cross-check.

We cross-checked the equations and text. The Equation in line 47 of the original submission is correct for an upwind lidar beam (as the one we show in the figure). The equation $v_r = u \cos\phi + w \sin\phi$ is correct for the downwind case or for the case with 'negative' $\phi$ angles as we mentioned in the original submission in lines 54–55. To better clarify this, we now add "i.e., pointing upwind," just before the above equation is mentioned and add "or when the lidar points downwind" after "For the cases with negative $\phi$ values" later in the paragraph.

4. L64 – "...eddies with most energy". Actually, the energy-containing eddies are those at larger scales, larger than those belonging to the inertial subrange. I would rather say a probe volume small enough to probe turbulence processes at small scales, ideally down to those responsible for dissipation, which are proportional to the Kolmogorov scale.

Based on the comment of the reviewer, we have rephrased the sentence.

5. L197 – Can you provide references about the relationship between turbulence intensity and height? How much does it differ from other theories, e.g. Townsend wall-attached eddy hypothesis where $\sigma_u^2 = B_1 - A_1 \log \frac{z}{\delta}$?

As suggested, we now provide the reference to similarity theory [Stull, 1988], which is the basis of the approximation

6. L 317 – In Fig. 4 – make sure y-axes and x-axes have the same scale

Changed as suggested.

7. Conclusions: You should provide an overall summary of work describing the strategy, the objectives, not only the results. Please rework on the conclusions.

As suggested, we now add a summary of the work and the ideas before presenting the results

**References**

R. B. Stull. *An introduction to boundary layer Meteorology*. Kluwer Academic Publishers, 1988.